# BQ-NCO: Bisimulation Quotienting for Generalizable Neural Combinatorial Optimization

## Abstract

Despite the success of Neural Combinatorial Optimization methods for end-to-end heuristic learning, out-of-distribution generalization remains a challenge. In this paper, we present a novel formulation of combinatorial optimization (CO) problems as Markov Decision Processes (MDPs) that effectively leverages symmetries of the CO problems to improve out-of-distribution robustness. Starting from the standard MDP formulation of constructive heuristics, we introduce a generic transformation based on bisimulation quotienting (BQ) in MDPs. This transformation allows to reduce the state space by accounting for the intrinsic symmetries of the CO problem and facilitates the MDP solving. We illustrate our approach on the Traveling Salesman, Capacitated Vehicle Routing and Knapsack Problems. We present a BQ reformulation of these problems and introduce a simple attention-based policy network that we train by imitation of (near) optimal solutions for small instances from a single distribution. We obtain new state-of-the-art generalization results for instances with up to 1000 nodes from synthetic and realistic benchmarks that vary both in size and node distributions.

## 1 Introduction

Combinatorial Optimization problems are crucial in many application domains such as transportation, energy, logistics, etc. Because they are generally NP-hard (Cook et al., 1997), their resolution at real-life scales is mainly done by heuristics, which are efficient algorithms that generally produce good quality solutions (Boussaïd et al., 2013). However, strong heuristics are generally problem-specific and designed by domain experts. Neural Combinatorial Optimization (NCO) is a relatively recent line of research that focuses on using deep neural networks to learn such heuristics from data, possibly exploiting information on the specific distribution of problem instances of interest (Bengio et al., 2021; Cappart et al., 2021). Despite the impressive progress in this field over the last few years, their out-of-distribution generalization, especially to larger instances, remains a major hurdle (Joshi et al., 2022; Manchanda et al., 2022).

In this paper, we are interested in constructive NCO methods, which build a solution incrementally, by applying a sequence of elementary steps. These methods are often quite generic, see e.g. the seminal papers by Khalil et al. (2017); Kool et al. (2019). Most CO problems can indeed be represented in this way, although the representation is not unique as the nature of the steps is, to a large extent, a matter of choice. Given a choice of step space, solving the CO problem amounts to computing an optimal policy for sequentially selecting the steps in the construction. This task can typically be performed in the framework of Markov Decision Processes (MDP), through imitation or reinforcement learning. The exponential size of the state space, inherent to the NP-hardness of combinatorial problems, usually precludes other methods such as (tabular) dynamic programming. Whatever the learning method used to solve the MDP, its efficiency, and in particular its out-of-distribution generalization capabilities, greatly depends on the state representation. The state space is often characterized by deep symmetries, which, if they are not adequately identified and leveraged, hinders the training process by forcing it to independently learn the policy at states which in fact are essentially the same (modulo some symmetry).

In this work, we investigate a type of symmetries which often occurs in MDP formulations of constructive CO heuristics. We first introduce a generic framework to systematically derive a naive CO problem-specific MDP. We formally demonstrate the equivalence between solving the MDP and

solving the CO problem and highlight the flexibility of the MDP formulation, by defining a minimal set of conditions for the equivalence to hold. Our framework is general and easy to specialize to encompass previously proposed learning-based construction heuristics. We then show that the state space of this naive MDP is inefficient because it fails to capture deep symmetries of the CO problem, even though such symmetries are easy to identify. Therefore, we propose a method to transform the naive MDP, based on the concept of *bisimulation quotienting* (BQ), in order to get a reduced state space, which is easier to process by the usual (approximate) MDP solvers. We illustrate our approach on three well-known CO problems, the Traveling Salesman Problem (TSP), the Capacitated Vehicle Routing Problem (CVRP) and Knapsack Problem (KP). Furthermore, we propose a simple transformer-based architecture for these problems, that we train by imitation of expert trajectories derived from (near) optimal solutions. In particular, we show that our model is well-suited for our BQ formulation: it spends a monotonically increasing amount of computation as a function of the subproblem size (and therefore complexity), in contrast to most previous models. Finally, extensive experiments confirm the validity of our approach, and in particular its state-of-the-art out-of-distribution generalization capacity. In summary, our contributions are as follows: 1) We present a generic and flexible framework to define a construction heuristic MDP for arbitrary CO problems; 2) We propose a method to simplify commonly used "naive" MDPs for constructive NCO via symmetry-focused bisimulation quotienting; 3) We design an adequate transformer-based architecture for the new MDP, for the TSP, CVRP and KP; 4) We achieve state-of-the-art generalization performance on these three problems.

## 2    COMBINATORIAL OPTIMIZATION AS A MARKOV DECISION PROBLEM

In this section, we present a generic formalization of constructive heuristics which underlies their MDP formulation. A deterministic CO problem is denoted by a pair $(\mathcal{F}, \mathcal{X})$, where $\mathcal{F}$ is its objective function space and $\mathcal{X}$ its (discrete) solution space. A problem instance $f \in \mathcal{F}$ is a mapping $f : \mathcal{X} \rightarrow \mathbb{R} \cup \{\infty\}$, with the convention that $f(x) = \infty$ if $x$ is infeasible for instance $f$. A solver for problem $(\mathcal{F}, \mathcal{X})$ is a functional:

$$\text{SOLVE} : \mathcal{F} \rightarrow \mathcal{X} \qquad \text{satisfying} \qquad \text{SOLVE}(f) = \arg\min_{x \in \mathcal{X}} f(x). \qquad (1)$$

**Incremental solution construction**   Constructive heuristics for CO problems build a solution sequentially, starting at an empty partial solution and expanding it at each step until a finalized solution is reached. Many NCO approaches are based on a formalization of that process as an MDP, e.g. Khalil et al. (2017); Kool et al. (2019); Zhang et al. (2020). Such an MDP can be obtained, for an arbitrary CO problem $(\mathcal{F}, \mathcal{X})$, using the following ingredients:

• **Steps**: $\mathcal{T}$ is a set of available steps to construct solutions. A *partial solution* is a pair $(f, t_{1:n})$ of a problem instance $f \in \mathcal{F}$ and a sequence of steps $t_{1:n} \in \mathcal{T}^*$ (the set of sequences of elements of $\mathcal{T}$). Observe that a partial solution (in $\mathcal{F} \times \mathcal{T}^*$) is *not* a solution (in $\mathcal{X}$), but may *represent* one.

• **Representation**: $\text{SOL} : \mathcal{F} \times \mathcal{T}^* \rightarrow \mathcal{X} \cup \{\bot\}$ is a mapping that takes a partial solution and returns either a feasible solution (in which case the partial solution is said to be finalized), or $\bot$ otherwise.

• **Evaluation**: $\text{VAL} : \mathcal{F} \times \mathcal{T}^* \rightarrow \mathbb{R} \cup \{\infty\}$ is a mapping that takes a partial solution and returns an estimate of the minimum value of its expansions into finalized solutions. If the returned value is finite, the partial solution is said to be *admissible*.

In order to define an MDP using these ingredients, we assume they satisfy the following axioms:

$$\forall f \in \mathcal{F}, x \in \mathcal{X}, \quad f(x) < \infty \iff \exists t_{1:n} \in \mathcal{T}^* \text{ such that } \text{SOL}(f, t_{1:n}) = x, \qquad (2a)$$

$$\forall f \in \mathcal{F}, t_{1:n} \in \mathcal{T}^*, \quad \text{SOL}(f, t_{1:n}) \neq \bot \implies \forall m \in \{1:n-1\}, \text{SOL}(f, t_{1:m}) = \bot, \qquad (2b)$$

$$\forall f \in \mathcal{F}, t_{1:n} \in \mathcal{T}^*, x \in \mathcal{X}, \quad \text{SOL}(f, t_{1:n}) = x \implies \begin{cases} \text{VAL}(f, t_{1:n}) = f(x), \\ \forall m \in \{1:n-1\}, \text{VAL}(f, t_{1:m}) < \infty. \end{cases} \qquad (2c)$$

Equation 2a states that the feasible solutions are exactly those represented by a finalized partial solution; Equation 2b states that if a partial solution is finalized then none of its preceding partial solutions in the construction can also be finalized; Equation 2c states that the evaluation of a finalized partial solution is the value of the solution it represents, and all its preceding partial solutions are admissible.

We call a triplet $\langle \mathcal{T}, \text{SOL}, \text{VAL} \rangle$ satisfying the above axioms a *specification* of problem $(\mathcal{F}, \mathcal{X})$. Note that a specification is not intrinsic to the problem. The step space $\mathcal{T}$ results from a choice

of how to construct a solution sequentially. Once $\mathcal{T}$ is chosen, SOL is determined, and must satisfy Equation 2a and 2b. Then VAL is only loosely constrained by Equation 2c, and can be chosen among a wide range of alternatives, including the following straightforward, uninformed one and the ideal, but intractable one (and, more likely, somewhere in between these two extremes):

$$\text{VAL}^{\text{uninformed}}(f, t_{1:n}) =_{\text{def}} f(x) \text{ if } [\text{SOL}(f, t_{1:n}) = x \neq \perp] \text{ else } 0,$$

$$\text{VAL}^{\text{ideal}}(f, t_{1:n}) =_{\text{def}} \min\{f(x) | x \in \mathcal{X}, \exists u_{1:m} \in \mathcal{T}^* \text{ s.t. } \text{SOL}(f, t_{1:n} u_{1:m}) = x\},$$

with the convention $\min \emptyset = \infty$. Value 0 in the uninformed case can be replaced by any constant.

**Solution construction as an MDP**   Using a specification $\langle \mathcal{T}, \text{SOL}, \text{VAL} \rangle$ of problem $(\mathcal{F}, \mathcal{X})$, one can derive a "naive" MDP as follows. **States** are partial solutions (in $\mathcal{F} \times \mathcal{T}^*$); **actions** are steps (in $\mathcal{T}$); a state is **terminal** if it is a finalized partial solution; **transitions**: action $u \in \mathcal{T}$ applied to a non-terminal state $(f, t_{1:n})$ leads to state $(f, t_{1:n}u)$ where $u$ is appended to the sequence so far, with **reward** $\text{VAL}(f, t_{1:n}) - \text{VAL}(f, t_{1:n}u)$, conditioned on $\text{VAL}(f, t_{1:n}u)$ being finite. Note that VAL has the double role of providing a reward and specifying the set of allowed actions. The number of these is expected to be linear, or at worst polynomial, in the size of the instance, since picking a step should not be as complex as solving the whole problem.

Now, assume we have access to a generic solver $\text{SOLVE}^{\text{MDP}}$, which, given an MDP $\mathcal{M}$ and one of its states $s_o$, returns an optimal trajectory starting at that state, i.e. $\arg \max_\tau R(\tau)$ where $\tau$ ranges over the $\mathcal{M}$-trajectories starting at $s_o$ and ending in a terminal state, and $R(\tau)$ denotes its cumulated reward. Note that because we are dealing with deterministic MDPs, looking for an optimal policy is the same as looking for an optimal trajectory for a given set of initial states. That is why $\text{SOLVE}^{\text{MDP}}$ is defined here directly in terms of trajectories rather than policies. $\text{SOLVE}^{\text{MDP}}$ can then be specialized into a solver for the specific CO problem $(\mathcal{F}, \mathcal{X})$:

**Proposition 1.** *Let $\mathcal{M}_o$ be the naive MDP obtained from specification $\langle \mathcal{T}, \text{SOL}, \text{VAL} \rangle$. The procedure defined as follows (where $\epsilon$ denotes the empty sequence) satisfies the requirement of equation 1:*

$$\text{SOLVE}(f \in \mathcal{F}) =_{\text{def}} \{\text{SOL}(s) | s \text{ is the last state of the trajectory } \text{SOLVE}^{\text{MDP}}(\mathcal{M}_o, (f, \epsilon))\}.$$

In other words, solving the naive MDP is equivalent to solving the CO problem. The detailed proof of Proposition 1 is in Appendix F. Of course, procedure $\text{SOLVE}^{\text{MDP}}$ may be approximate, in which case so is procedure SOLVE. Moreover, its performance depends on that of $\text{SOLVE}^{\text{MDP}}$, esp. its out-of-distribution generalization capacity, but also on the choice of specification, esp. of action space. It is a distinguishing feature of CO from an MDP perspective that the action space is *not* prescribed by the problem.

The impact of the choice of the VAL mapping depends on the type of learning used by $\text{SOLVE}^{\text{MDP}}$. When $\text{SOLVE}^{\text{MDP}}$ learns by *reinforcement*, VAL is essential, as it provides the rewards which guide the resolution. For example, $\text{VAL}^{\text{uninformed}}$ leads to the notoriously hard case of sparse rewards, while $\text{VAL}^{\text{ideal}}$ (were it tractable) would lead to the trivial case where a myopic policy (greedy in its immediate reward) is optimal. Although we do not provide a generic method to design VAL, we argue that there are natural candidates, typically based on extending the objective function to partial solutions (not just finalized ones). When $\text{SOLVE}^{\text{MDP}}$ learns by *imitation* instead, the choice of VAL has a much more limited impact: it only serves to define the allowed actions. The critical factor in that case is the construction of the training dataset of expert trajectories to imitate.

**Example on TSP**   Consider the widespread CO problem known as the Traveling Salesman Problem (TSP) in a Euclidian space $V$. A TSP solution (in $\mathcal{X}$) is a path, i.e. a finite sequence of pairwise distinct nodes. A TSP instance (in $\mathcal{F}$) is given by a finite set $D$ of nodes as points in $V$, and maps any solution (path) to the length of that path (closed at its ends) if it visits exactly all the nodes of $D$, and $\infty$ otherwise (infeasible solutions).

A simple specification $\langle \mathcal{T}, \text{SOL}, \text{VAL} \rangle$ for the TSP is given by: the step space $\mathcal{T}$ is the set of nodes; for a given instance $f$ and sequence $t_{1:n}$ of steps, $\text{SOL}(f, t_{1:n})$ is either the sequence $t_{1:n}$ if it forms a path which visits exactly all the nodes of $f$, or $\perp$ otherwise; and $\text{VAL}(f, t_{1:n})$ is either the length of path $t_{1:n}$ (closed at its ends) if it forms a path which visits only nodes of $f$ (maybe not all), or $\infty$ otherwise. It is easy to show that we thus obtain a specification (as defined by the axioms above) of the TSP. In TSP-MDP, the naive MDP obtained from it, the reward of taking action $u$ (a node)

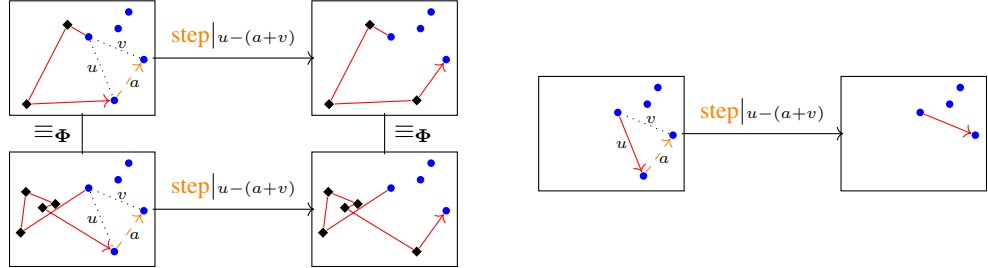

Figure 1: An example of bisimulation commutation in TSP-MDP, and the corresponding path-TSP-MDP transition. The step is the same in all three transitions: it is the end node of the dashed arrow. And the reward is also the same: it depends only on the distances $a, u, v$, and not on any of the previously visited nodes.

at state $(f, t_{1:n})$ is $\delta_f(t_n, t_1) - (\delta_f(t_n, u) + \delta_f(u, t_1))$ where $\delta_f$ is the node distance measured on the corresponding points of $V$ in $f$, conditioned on $t_{1:n}u$ being pairwise distinct nodes of $f$. Observe that when allowed, the reward depends only on the start and end nodes $t_1, t_n$ of the step sequence.

## 3 BISIMULATION QUOTIENTING FOR COMBINATORIAL OPTIMIZATION

In our context of deterministic CO problems and therefore deterministic MDPs, the general notion of *bisimilarity* is simplified (Givan et al., 2003): two states are said to be bisimilar if they spawn exactly the same action-reward sequences. Likewise, the notion of a binary relation $\mathcal{R}$ on states being a *bisimulation* reduces to a commutation between the (deterministic) transitions of the MDP and that relation: if $s_1 \mathcal{R} s_2$ and action $a$ applied to state $s_1$ leads to state $s_1'$ with reward $r$, then action $a$ applied to state $s_2$ leads to a state $s_2'$ with the same reward $r$, and $s_1' \mathcal{R} s_2'$. An illustration is given in Fig 1. Bisimilarity is equivalently defined as the largest bisimulation (see Appendix H.1).

**Bisimilarity-induced symmetries**   In the naive MDP obtained from a specification of a given CO problem, a state is a partial solution and carries the whole information about the "past" decisions (steps) leading to it, which may not all be useful for the "future" decisions, i.e. the completion of that partial solution. Consider for example the following two states in TSP-MDP, in which the sequence of steps of the partial solution is represented as a directed path in red among some of the problem instance nodes:

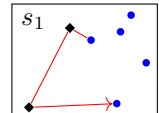 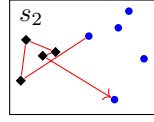

Observe that $s_1, s_2$ differ only in the inner nodes of the red path (black diamond-shaped nodes). Now, it is easy to see that the successful completions of these two partial solutions are identical: they each consist of a path visiting the (same) unvisited (blue) nodes, starting at the end node of the red path and ending at its start node, with the same rewards defined by VAL. Consequently, in the MDP, the two states $s_1, s_2$ spawn exactly the same action-reward sequences and form a *bisimilar* pair. This is the kind of deep symmetries of the problem which we want the MDP to leverage. Of course, there exist other kinds of symmetries, e.g. rotational symmetries: if $s_2$ is obtained from $s_1$ by applying an isometric transformation to all the points in the problem instance, then $s_1, s_2$ also form a bisimilar pair. However, the latter symmetry is specific to the Euclidian version of the TSP. We focus here on the former kind of symmetry as it is more general. Although it has previously been noted for routing problems (Peng et al., 2020; Xin et al., 2021c), we show here that it is an inherent characteristic of constructive CO approaches.

**Bisimulation quotienting**   A classical result on MDPs (Givan et al., 2003) states that *all* such symmetries in *any* MDP can be leveraged by *quotienting* it by its bisimilarity relation, i.e. the set of all bisimilar pairs. Of course, there is no free lunch: constructing the bisimilarity of an MDP is in

general intractable. Still, the result remains valuable because it holds for any bisimulation, not just the bisimilarity. Therefore one can control the amount of symmetries captured in the quotienting by carefully choosing the bisimulation, trading off its closeness to full bisimilarity for tractability.

We now assume that, for a given CO problem $(\mathcal{F}, \mathcal{X})$ we have access not only to a specification $\langle \mathcal{T}, \text{SOL}, \text{VAL} \rangle$ with its associated naive MDP, but also to a mapping $\mathbf{\Phi}:\mathcal{F}\times\mathcal{T}^*\to\hat{\mathcal{S}}$ from partial solutions to some new space $\hat{\mathcal{S}}$. Typically, $\mathbf{\Phi}(f, t_{1:n})$ should capture, within the partial solution $(f, t_{1:n})$, a piece of information as small as possible but sufficient to determine the set of action-reward sequences it spawns in the MDP – in other words, a summary of its "past" which is sufficient to determine its "future". We can then define an equivalence relation $\equiv_{\mathbf{\Phi}}$ where two partial solutions are equivalent if they have the same image by $\mathbf{\Phi}$. For it to be a bisimulation, $\mathbf{\Phi}$ must satisfy:

$$\forall s_1, s_2 \in \mathcal{F}\times\mathcal{T}^*, \quad \mathbf{\Phi}(s_1)=\mathbf{\Phi}(s_2) \Rightarrow \left\{ \begin{array}{l} \forall u\in\mathcal{T}, \ \mathbf{\Phi}(s_1 u)=\mathbf{\Phi}(s_2 u) \\ \text{and } \text{VAL}(s_1)-\text{VAL}(s_1 u)=\text{VAL}(s_2)-\text{VAL}(s_2 u). \end{array} \right. \tag{3}$$

Under that assumption, we can construct a new MDP (the quotient of the original one by the bisimulation) which is equivalent, as far as policy optimization is concerned, to the original one, but captures more symmetries of the problem. This allows to reduce the state space and should lead to a better performance, whatever the generic MDP solver used afterwards. Furthermore, by construction, the equivalence classes are in one-to-one correspondence with the states in $\hat{\mathcal{S}}$, so that the new MDP can be formulated on that space directly.

**Application to the TSP, CVRP and KP** Let $\mathbf{\Phi}$ be the mappings from TSP-MDP states (TSP states for short) into new objects called "path-TSP" states, informally described by the following diagram:

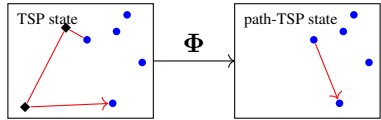

The inner nodes (black diamonds) on the red path of visited nodes in the TSP state are removed, leaving only the two ends of the red path which constitute two distinguished nodes in the path-TSP state. Mapping $\mathbf{\Phi}$ has been designed to satisfy equation 3, so it induces a bisimulation on TSP-MDP (see Figure 1), and TSP-MDP can be turned into an equivalent "path-TSP-MDP" on path-TSP states. This path-TSP-MDP can be viewed as solving a variant of the TSP known as path-TSP, hence its name. However it is *not* the naive MDP for that variant since it forgets as it progresses, while naive MDPs always accumulate.

With the CVRP, we define a step as the pair of a node and a binary flag specifying whether that node is reached via the depot or directly. We can define a mapping $\mathbf{\Phi}$ similarly to the TSP case, except it is not sufficient to summarize the "past" (the visited nodes) by just the two ends of their path: to guarantee equation 3 and the bisimulation property, an additional piece of information must be preserved from the past, namely the remaining capacity at the end of the current path. For the KP, intuitively, the summary of the "past" is captured by the remaining items and the remaining knapsack capacity. This idea can be leveraged to design a bisimulation. Formal descriptions of the specifications and bisimulation quotienting for the CVRP and KP are provided in Appendices A and B, respectively.

## 4 NEURAL ARCHITECTURE FOR PATH-TSP

**Model** We now describe our proposed policy network for the path-TSP-MDP above. Figure 4 (Appendix) provides a quick overview. The models for path-CVRP and BQ-KP differ only slightly and are presented in Appendix A and B. Most neural models for TSP utilize an encoder-decoder architecture, in which the encoder computes a representation of the entire graph once, and the decoder constructs a solution by taking into consideration the representation of the whole graph and the partial solution, e.g. Attention Model (Kool et al., 2019), or PointerNetworks (Vinyals et al., 2015). In our case, the path-TSP formulation allows us to forget the nodes in the graph that have already been visited, except the distinguished origin and destination nodes. As a corollary,

it also requires re-encoding the remaining nodes at each prediction step – hence removing the need for a separate auto-regressive decoder. To encode a path-TSP state, we use a Transformer model (Vaswani et al., 2017). Each node is represented by its $(x, y)$ coordinates, so that the input feature matrix for an $N$-node state is an $N \times 2$ matrix. We embed these features via a linear layer. The remainder of the encoder is based on Vaswani et al. (2017) with the following differences. First, we do not use positional encoding since the input nodes have no order. Instead, we learn an origin (resp. destination) embedding that is added to the feature embedding of the origin (resp. destination) node. Second, we use ReZero (Bachlechner et al., 2021) normalization, which leads to more stable training and better performance in our experiments (see ablation study in Appendix D). Finally, a last linear layer projects the encoder's output into a vector of size $N$, from which unfeasible actions, corresponding to the origin and destination nodes, are masked out, before applying a softmax operator so as to interpret the scalar node values for all allowed nodes as action probabilities.

**Training** We train our model by imitation of expert trajectories, using a plain cross-entropy loss (behaviour cloning). Such trajectories are extracted from pre-computed optimal (or near optimal) solutions for instances of a (relatively small) fixed size. Note that (optimal) solutions are not directly in the form of trajectories. Equation 2a guarantees that a trajectory exists for any solution, but it is usually far from unique. Besides, optimal solutions are costly, so we seek to make the most out of each of them. In the TSP case, we observe that given an optimal tour, any sub-path of that tour is also an optimal solution to the associated path-TSP sub-problem, hence amenable to our path-TSP model. We therefore form minibatches by first sampling a number $n$ between 4 and $N$ (path-TSP problems with less than 4 nodes are trivial), then sampling sub-paths of length $n$ – the same $n$ for all the minibatch entries so as to simplify batching – from the initial solution set. For the CVRP, the procedure is similar, except that, first, the extracted sub-paths must end at the depot, and, second, they can follow the sub-tours of the full solution in any order. We observed experimentally that the way that order is sampled has an impact on the performance (see Appendix E).

**Complexity** Because of the quadratic complexity of self-attention, and the fact that we call our model at each construction step, the total complexity is $\mathcal{O}(N^3)$[1] where $N$ is the instance size. Note that closely related Transformer-based models such as the TransformerTSP (Bresson & Laurent, 2021) and the Attention Model (Kool et al., 2019) have a total complexity of $\mathcal{O}(N^2)$[2] At each decision step, for $t$ remaining nodes, our model has a budget of $\mathcal{O}(t^2)$ compute whereas previous models only spend $\mathcal{O}(t)$. We believe that this is a useful inductive bias, which enables better generalization in particular for larger problem sizes. This hypothesis is supported by the fact that replacing the self-attention component with a linear time alternative (i.e., spending $\mathcal{O}(t)$ operations per step) drastically degrades the generalization ability to larger instances, as we show in Appendix D,

**Summary** By reformulating TSP-MDP into path-TSP-MDP, the state is made to contain only a very concise summary of the "past" of a partial solution (how it was formed) as two distinguished nodes, but sufficient to determine its "future" (how it can be completed). Furthermore, at train time, we sample optimal solutions and associated path-TSP states amongst all the possible infixes of solutions of full problems. These proposed modifications go hand-in-hand. Thanks to the transformation to path-TSP-MDP, our model enables better generalization in two important ways: (i) Due to re-encoding at each step, the encoder produces a graph representation that is specific to the current path-TSP-MDP state. Graphs in these states vary in size and distribution, implicitly encouraging the model to work well across sizes and node distributions, and generalize better than if such variations were not seen during the training. In this regard, our model is similar to the SW-AM model (Xin et al., 2021c), except that they only approximate the re-embedding process in practice. (ii) By sampling subsequences from our training instances, we automatically get an augmented dataset, which some previous models had to explicitly design their model for (Kwon et al., 2021).

---

[1] More precisely, the complexity is proportional to $\sum_{t=1}^{N} t^2 = N(N+1)(2N+1)/6$ hence the $\mathcal{O}(N^3)$.
[2] After an encoder of complexity $\mathcal{O}(N^2)$, the decoder has linear complexity $\mathcal{O}(N-t)$ at step $t$.

## 5 RELATED WORK

**NCO approaches** Many NCO approaches construct solutions sequentially, via auto-regressive models. Starting with the seminal work by Vinyals et al. (2015), which proposed the Pointer network that was based on RNNs and trained in a supervised way, Bello et al. (2017) trained the same model by RL for the TSP and Nazari et al. (2018) adapted it for the CVRP. Kool et al. (2019) introduced an attention-based encoder-decoder architecture (AM) trained with RL to solve several variants of routing problems – which is reused by Kwon et al. (2021) along with a few extensions (POMO). TransformerTSP Bresson & Laurent (2021) use a similar architecture with a different decoder on TSP problems. Another line of works is concerned with directly producing a heat-map of solution segments: Nowak et al. (2018) trained a Graph Neural Network in a supervised manner to output an adjacency matrix, which is converted into a feasible solution using beam search. Joshi et al. (2019) followed a similar framework and trained a deep Graph Convolutional Network instead, that was used by (Fu et al., 2020) as well.

**Step-wise methods** Peng et al. (2020) first pointed out the limitation of the original AM (Kool et al., 2019) approach in representing the dynamic nature of routing problems. They proposed to update the encoding after each subtour completion for the CVRP. Xin et al. (2021c) proposed a similar step-wise strategy but the encodings recomputed after each decision. In practice, their architecture is the most similar to ours for the TSP. However, thanks to our principled MDP transformations based on bisimulation quotienting, we obtain a superior representation for CVRP: In contrast to our approach, their CVRP architecture only provides censored information by omitting the remaining vehicle capacity and simply restricting the state to the nodes whose demand is below the remaining capacity. Xin et al. (2021b) extended on this idea by proposing the Multi-Decoder Attention Model (MDAM) that in particular contains a special layer to efficiently approximate the re-embedding process. As MDAM constitutes the most advanced version, we employ it as a baseline in our experiments.

**Generalizable NCO** Generalization to different instances distributions, and esp. larger instances, is regarded as one of the major limitations of current NCO approaches (Joshi et al., 2022; Mazyavkina et al., 2020). Fu et al. (2020) trained a Graph Convolution model in a supervised manner on small graphs and used it to solve large TSP instances, by applying the model on sampled subgraphs and using an expensive MCTS search to improve the final solution (Att-GCN+MCTS). While this method achieves excellent generalization on TSP instances, MCTS requires a lot of computing resources and is essentially a post-learning search strategy. Geisler et al. (2022) investigate the robustness of NCO solvers through adversarial attacks and find that existing neural solvers are highly non-robust to out-of-distribution examples. They conclude that one way to address this issue is through adversarial training. In particular, Xin et al. (2021a) trains a GAN to generate instances that are difficult to solve for the current model. Manchanda et al. (2022) take a different approach and leverage meta-learning to learn a model in such a way that it is easily adaptable to new distributions. Accounting for symmetries in a given CO problem is a powerful idea to boost the generalization performance of neural solvers. Both Kwon et al. (2021) and Kim et al. (2022) make use of solution symmetricity as part of their loss function during training. Problem instance symmetry can be used at training time to augment the dataset (Kwon et al., 2021) or enforce robust representations (Kim et al., 2022), or it can be used at inference time to augment the set of solutions (Kwon et al., 2021).

Please note that all of the above are orthogonal to our approach: rather than augmenting data or changing the training paradigm, our approach simplifies the state space by transforming the MDP, which has beneficial effects irrespective of the method of training.

## 6 EXPERIMENTS

To verify the effectiveness of our method, we test it on TSP, CVRP and KP. This section presents experimental results for TSP and CVRP, while results for KP are presented in Appendix B.

We train our model and all baselines on synthetic TSP and CVRP instances of size 100, generated as in Kool et al. (2019). We choose graphs of size 100 because it is the largest size for which (near) optimal solutions are still reasonably fast to obtain, and such training datasets are commonly used in the literature. Then, we evaluate trained models on synthetic instances of size 100, 200, 500 and 1K generated from the same distribution, as well as the standard TSPLib and CVRPLib datasets.

**Hyperparameters and training procedure** We use the same hyperparameters for all problems. The model has 9 layers, each built with 8 attention heads with embedding size of 128 and dimension of feed-forward layer of 512. Our model is trained on 1 million instances with 100 nodes split into batches of size 1024, for 1000 epochs. Solutions of these problems are obtained by using the Concorde solver (Applegate et al., 2015) for TSP and LKH heuristic (Helsgaun, 2017) for CVRP. We use Adam (Kingma & Ba, 2017) as optimizer with an initial learning rate of $7.5e-4$ and decay of 0.98 every 20 epochs.

**Evaluation** We compare our model with existing state-of-the-art methods: OR-Tools (Perron & Furnon, 2022), LKH (Helsgaun, 2017), and Hybrid Genetic Search (HGS) for the CVRP (Vidal, 2022) as traditional non-neural methods; Att-GCN+MCTS and NeuralRewriter (Chen & Tian, 2019) as hybrid methods for TSP and CVRP respectively; and deep learning-based constructive methods: AM, TransformerTSP, MDAM and POMO, which were discussed in Section 5. For all deep learning baselines we use the model *trained on graphs of size 100* and the best decoding strategy. Following the same procedure as in Fu et al. (2020), we generate four test datasets with graphs of sizes 100, 200, 500 and 1000. For CVRP, we use capacities of 50, 80, 100 and 250, respectively. In addition, we report the results on TSPLib instances with up to 4461 nodes and all CVRPLib instances with node coordinates in the Euclidian space. For all models, we report the optimality gap and the inference time. The optimality gap for TSP is based on the optimal solutions obtained with Concorde. For CVRP, although HGS gives better results than LKH, we use the LKH solution as a reference to compute the "optimality" gap, in order to be consistent (and easily comparable) with previous works. While the optimality gap is easy to compute and compare, measurements of running times are much harder: they may vary due to the implementation platforms (Python, C++), hardware (GPU, CPU), parallelization, batch size, etc. Therefore, we also report the number of solutions generated by each of the constructive deep learning models. In our experiments, we run all deep learning models on a single Nvidia Tesla V100-S GPU with 24GB memory, and other solvers on Intel(R) Xeon(R) CPU E5-2670 with 256GB memory, in one thread.

**Results** Tables 1a and 1b summarize our results on TSP and CVRP, respectively. For both problems, our model shows superior generalization on larger graphs, even with the greedy decoding strategy, which generates only a single solution while all others generate several hundreds (and select the best among them). In terms of running time with greedy decoding, our model is competitive with the POMO baseline, and significantly faster than other models. Beam search decoding with beam size 16 further improves the quality of solutions, but as expected, it takes approximately 16 times longer. Figure 2 shows optimality gap versus running time for our model and other baseline models. Our model clearly outperforms other models in terms of generalization to larger instances. The only model that is competitive with ours is Att-GCN+MCTS, but it is 2-15 times slower and is designed for TSP only. In addition to synthetic datasets, we test our model on TSPLib and VRPLib instances, which are of varying graph sizes, node distributions, demand distributions and vehicle capacities. Table 1c shows our model's strength over MDAM and POMO, even with the greedy decoding strategy. The effectiveness of our MDP transformation method and the resulting neural architecture is confirmed by the results. Thanks to our more principled approach that leads to better state representations and a simpler architecture without a decoder, by generating a single solution, it is able to outperform MDAM (with 250 solutions), which is closest to our model conceptually. Moreover, an ablation study in Appendix D suggests that spending appropriate amounts of compute for each subproblem is a crucial factor in our model.

# 7 CONCLUSION

We have presented a flexible framework to derive MDPs that sequentially construct solutions to CO problems. Starting from a naive MDP, we introduced a generic transformation using bisimulation quotienting, which reduces the state space by leveraging its symmetries. We applied this transformation on the TSP and CVRP, for which we also designed a simple attention-based model, well-suited to the transformed state representation. We show experimentally that this combination of state representation, simple model, and training procedure yields state-of-the-art generalization results on diverse benchmarks. While training on relatively small instances allowed us to use imitation learning, our approach and model could be similarly used with reinforcement learning. Finally, we have focused on deterministic CO problems, leaving the adaptation of our framework to stochastic problems as future work.

Table 1: Summary of the experimental results. The bold values represent the best optimality gap (lower is better) and fastest inference time. The underlined cells represent the best ratio between the quality of the solution and the inference time. #s refers to number of generated solutions.

| | #s | TSP100 | | TSP200 | | TSP500 | | TSP1000 | |
|---|---|---|---|---|---|---|---|---|---|
| Concorde | - | 0.000% | 38m | 0.000% | 2m | 0.000% | 40m | 0.000% | 2.5h |
| OR-Tools | - | 3.765% | 1.1h | 4.516% | 4m | 4.891% | 31m | 5.021% | 2.4h |
| Att-GCN+MCTS* | - | 0.037% | 15m | 0.884% | 2m | 2.536% | 6m | 3.223% | 13m |
| AM bs1024 | 1024 | 2.510% | 20m | 6.176% | 1m | 17.978% | 8m | 29.750% | 31m |
| TransTSP bs1024 | 1024 | 0.456% | 51m | 5.121% | 1m | 36.142% | 9m | 76.215% | 37m |
| MDAM bs50 | 250 | 0.395% | 45m | 2.044% | 3m | 9.878% | 13m | 19.965% | 1.1h |
| POMO augx8 | 8N | 0.134% | **1m** | 1.572% | **5s** | 20.182% | **1m** | 40.603% | 10m |
| BQ (ours) greedy | 1 | 0.540% | **1m** | 0.793% | **5s** | 1.425% | **1m** | 2.335% | **7m** |
| BQ (ours) bs16 | 16 | **0.032%** | 18m | **0.166%** | 1m | **0.682%** | 15m | **1.311%** | 1.8h |

(a) Experimental results on TSP. *We could not run Att-GCN+MCTS code on our architecture so we report results from the original paper.

| | #s | CVRP100 | | CVRP200 | | CVRP500 | | CVRP1000 | |
|---|---|---|---|---|---|---|---|---|---|
| LKH | - | 0.000% | 15.3h | 0.000% | 30m | 0.000% | 1.3h | 0.000% | 2.8h |
| HGS | - | -0.510% | 15.3h | -1.024% | 30m | -1.252% | 1.3h | -1.104% | 2.8h |
| OR-Tools | - | 9.617% | 15.3h | 10.700% | 30m | 11.403% | 1.3h | 13.559% | 2.8h |
| NeuRewriter * | - | 3.456% | 1.1h | 29.460% | 9m | 25.051% | 32m | 29.542% | 1.8h |
| AM bs1024 | 1024 | 4.180% | 24m | 7.786% | 1m | 16.964% | 8m | 86.410% | 31m |
| MDAM bs50 | 250 | 2.206% | 56m | 4.332% | 3m | 9.994% | 14m | 28.015% | 1.4h |
| POMO augx8 | 8N | **0.689%** | **1m** | 4.767% | **5s** | 20.575% | **1m** | 141.058% | 10m |
| BQ (ours) greedy | 1 | 4.832% | **1m** | 3.723% | **5s** | 3.429% | **1m** | 6.809% | **7m** |
| BQ (ours) bs16 | 16 | 1.798% | 18m | **1.375%** | 1m | **0.817%** | 15m | **2.048%** | 1.8h |

(b) Experimental results on CVRP. *We could not reproduce the reported results for NeuRewriter, so for CVRP100 we report results from the original paper and for other sizes we report the best result we got.

| | MDAM | POMO | BQ (ours) | |
|---|---|---|---|---|
| Size | bs50 | x8 | greedy | bs16 |
| <100 | 3.06% | 0.42% | 0.38% | **0.06%** |
| 100-200 | 5.14% | 2.31% | 2.82% | **1.61%** |
| 200-500 | 11.32% | 13.32% | 3.31% | **2.07%** |
| 500-1K | 20.40% | 31.58% | 10.08% | **3.04%** |
| >1K | 40.81% | 62.61% | 11.87% | **8.61%** |
| All | 19.01% | 26.30% | 6.22% | **3.94%** |

| | MDAM | POMO | BQ (ours) | |
|---|---|---|---|---|
| Set (size) | bs50 | augx8 | greedy | bs16 |
| A (32-80) | 6.17% | 4.86% | 5.85% | **1.96%** |
| B (30-77) | 8.77% | 5.13% | 7.04% | **3.50%** |
| F (44-134) | 16.96% | 15.49% | 7.20% | **3.04%** |
| M (100-200) | 5.92% | 4.99% | 6.69% | **1.85%** |
| P (15-100) | 8.44% | 14.69% | 4.71% | **1.32%** |
| X (100-1K) | 34.17% | 21.62% | 10.74% | **8.35%** |
| All (15-1K) | 22.36% | 15.58% | 8.58% | **5.60%** |

(c) Experimental results on TSPLib (left) and CVRPLib (right).

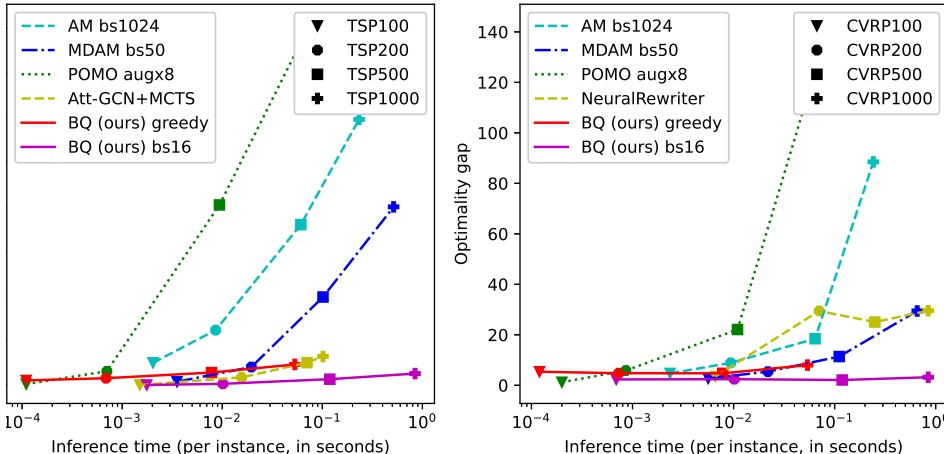

Figure 2: Generalization results on different graph sizes for TSP (left) and CVRP (right). Lower and further left is better.

## REPRODUCIBILITY STATEMENT

In order to ensure the reproducibility of our approach, we have:

- described precisely our generic theoretical framework (Section 2) and provided a detailed proof of Proposition 1 in Appendix F. This should in particular serve to adapt the framework to other CO problems;
- explained in detail our proposed model (Section 4 for TSP and Appendix A for CVRP), described precisely the training procedure and listed the hyperparameters (Section 6);
- used public datasets referenced in Section 6.

Furthermore, we plan to make our code public upon acceptance.

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

## A  APPLICATION TO THE CVRP

**Problem definition and specification**  The Capacitated Vehicle Routing Problem (CVRP) is a vehicle routing problem in which a vehicle (here, a single one) with limited capacity must deliver items from a depot location to various customer locations. Each customer has an associated *demand*, which represents an amount of items, and the problem is for the vehicle to provide all the customers in the least travel distance, returning as many times as needed to the depot to refill, but without ever exceeding the vehicle capacity.

Formally, we assume given a set of customer nodes, each with a demand (positive scalar), plus a depot node. A CVRP solution (in $\mathcal{X}$) is a finite sequence of nodes starting at the depot, which are pairwise distinct except for the depot, and respecting the capacity constraint: the total demand of any contiguous sub-sequence of customer nodes is below the vehicle capacity. A CVRP instance (in $\mathcal{F}$) is given by a finite set $D$ of nodes, including the depot, their coordinates in the Euclidian space $V$, and maps any solution to the length of the corresponding path using the distances in $V$, if the path visits exactly all the nodes of $D$, or $\infty$ otherwise (unfeasible solutions).

A possible specification $\langle \mathcal{T}, \text{SOL}, \text{VAL} \rangle$ for the CVRP is defined as follows. The step space $\mathcal{T}$ is the set of pairs of a non depot node and a binary flag indicating whether that node is to be reached via the depot or directly. The extension $\bar{t}$ of a step $t$ is either the singleton of its node component if its flag is 0 or the pair of the depot node and its node component if its flag is 1. For a given problem instance $f$ and sequence $t_{1:n}$ of steps, $\text{SOL}(f, t_{1:n})$ is either the sequence $\bar{t}_{1:n}$ if it forms a d-path which visits exactly all the nodes of $f$, or $\bot$ otherwise. $\text{VAL}(f, t_{1:n})$ is either the total length of $\bar{t}_{1:n}$ (closed at its end) if it forms a d-path which visits only nodes of $f$ (maybe not all), or $\infty$ otherwise. It is easy to show that $\langle \mathcal{T}, \text{SOL}, \text{VAL} \rangle$ forms a specification for the CVRP (i.e. satisfies the axioms of specifications introduced in Section 2). The naive MDP obtained from it is denoted CVRP-MDP.

**Bisimulation quotienting**  Just as with TSP, we can define a mapping $\Phi$ from CVRP-MDP states to a new "path-CVRP" state space, informally described by the following diagram.

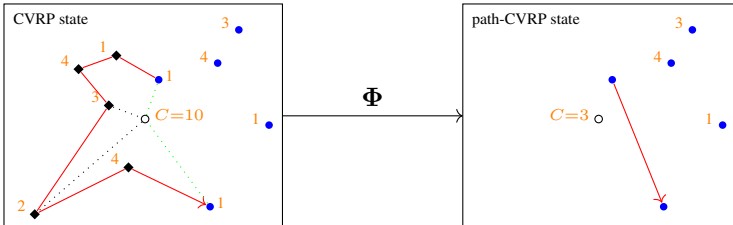

Here, the capacity of the vehicle is $C=10$, shown next to the (colourless) depot node, and the demand of each node is shown next to it, in orange. The black dotted line indicates that the action which introduced the node with demand 2 was via the depot: its flag was set to 1 (all the other actions had their flag set to 0 in this simple example). The green dotted line indicates how the path is closed to measure its length. After the node with demand 2, the path of visited nodes (in red) continues with nodes with demand 4 and 1, respectively, so that the remaining capacity at the end of the path is $C-(2+4+1)=3$. Compared to TSP, this is the additional piece of information in the summary of the "past" (path of visited nodes) which is preserved in the path-CVRP state, together with the origin and destination of the path. Mapping $\Phi$ thus defined satisfies Equation 3, hence induces a bisimulation on CVRP-MDP states, and by quotienting, one obtains an MDP which can be defined directly on path-CVRP states.

**Model architecture for CVRP**  The model architecture for CVRP is almost the same as for TSP, with a slight difference in the input sequence and in the output layer. In the TSP model, the input to the node embedding layer for a $N$-node state is a $2 \times N$ matrix of coordinates. For CVRP, we use two additional channels: one for node demands, and one for the current vehicle capacity, repeated across all nodes. The demand is set to zero for the origin and destination nodes. We obtain a $4 \times N$ matrix of features, which is passed through a learned embedding layer. As for TSP, a learned origin-signalling (resp. destination-signalling) vector is added to the corresponding embeddings. The rest of the architecture, in the form of attention layers, is identical to TSP, until after the action scores projection layer. In the case of TSP, the projection layer returns a vector of $N$ scores, where each score, after

a softmax, represents the probability of choosing the node as the next step in the construction. In the case of CVRP, the model returns a matrix of scores of dimension $N \times 2$, corresponding to each possible actions (node-flag pair) and the softmax scopes over this whole matrix. As usual, a mask is always applied to unfeasible actions before the softmax operator: those which have higher demand than the remaining vehicle capacity, as well as the origin and destination nodes.

# B   APPLICATION TO THE KNAPSACK PROBLEM

**Problem definition and specification**   The Knapsack Problem (KP) is classical combinatorial optimization problem in which we need to pack items, with given values and weights, into a knapsack with a given capacity. The objective is to maximize the total value of packed items. Formally, we assume given a set of items, each with a value and weight. A KP solution (in $\mathcal{X}$) is a subset of the items which respects a capacity constraint ("c-subset"): total weight of the items of the subset must not exceed the knapsack capacity. A KP instance (in $\mathcal{F}$) is given by a finite set of $D$ items and maps any c-subset to the sum of values of its items.

A simple problem specification $\langle \mathcal{T}, \text{SOL}, \text{VAL} \rangle$ can be defined as follows. The step space $\mathcal{T}$ is equal to the set of items, . For a partial solution $(f, t_{1:n})$, if the selected items satisfy the capacity constraints and adding any of the remaining items results in an infeasible solution, then $\text{SOL}(f, t_{1:n})$ returns the subset of selected items; otherwise it returns $\perp$. Finally, $\text{VAL}(f, t_{1:n})$ is either the sum of the values of the items in $t_{1:n}$ if they satisfy the capacity constraint and $\infty$ otherwise. Similarly to the TSP and CVRP cases, it is easy to show that $\langle \mathcal{T}, \text{SOL}, \text{VAL} \rangle$ forms a specification for the KP. The naive MDP obtained from it is denoted MDP-KP.

**Bisimulation quotienting**   As it was the case for TSP and CVRP, we can define a mapping $\Phi$ from KP-MDP state to a new "BQ-KP" state space, informally described by the following diagram.

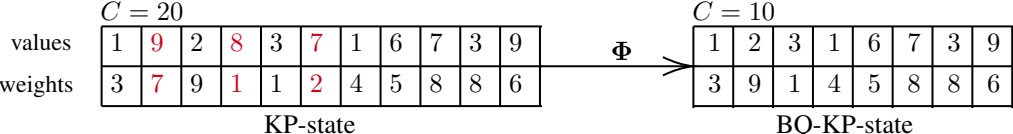

Here, capacity of the knapsack is $C = 20$ and each item is defined by its weight (bottom cell) and value (top cell). Mapping $\Phi$ for KP is straightforward - simply saying, it removes all picked items and update the remaining capacity by subtracting total weight of removed items from the previous capacity.

**Model architecture for KP**   The model architecture for KP is again very similar to previously described models for TSP and CVRP. The input to the model is a $3 \times N$ tensor composed of items properties (values, weights) and the additional channel for the remaining knapsack's capacity. By definition, the solution has no order (the result is a set of items), so there is no need to add tokens for origin and destination. A part from excluding these tokens and different input dimensions, the rest of the model is identical to the TSP model. The output is a vector of $N$ probabilities over all items with a mask over the unfeasible ones (with weights larger than remaining knapsack's capacity). In the training, at each construction step, any item of the ground-truth solution is a valid choice. Therefore we use a multi-class cross-entropy loss.

**Experimental results for KP**   We generate the training dataset as described in Kwon et al. (2021). We train our model on 1M KP instances of size 200 and capacity 25, with values and weights randomly sampled from the unit interval. We use the dynamic programming algorithm from ORTools to compute the ground-truth optinal solutions. As hyperparameters, we use the same as for the TSP. Then, we evaluate our model on test datasets with the number of items equal 200, 500 and 1000 and capacity of 25 and 50, for each problem size. Table B shows the performance of our model compared to POMO, one of the best performing NCO models on KP. Although our model does not outperform it in every dataset, it achieves better overall performance. It should be noted again that POMO builds $N$ solutions per instance and choose the best one, while our model generate a single solution per instance but still achieves better results.

|  |  | Optimal | POMO (single traj.) | | POMO (all traj.) | | BQ (greedy) | |
|---|---|---|---|---|---|---|---|---|
|  |  | value | value | opt gap | value | opt gap | value | opt gap |
| N=200 | C=25 | 58.023 | 57.740 | 0.476% | **58.007** | **0.017**% | 57.970 | 0.081% |
|  | C=50 | 80.756 | 79.483 | 1.544% | 79.787 | 1.170% | **80.710** | **0.056**% |
| N=500 | C=25 | 90.986 | 85.309 | 6.217% | 86.516 | 4.897% | **90.150** | **0.904**% |
|  | C=50 | 129.326 | 128.950 | 0.291% | **129.272** | **0.042**% | 128.369 | 0.739% |
| N=1000 | C=25 | 128.692 | 120.757 | 5.386% | **123.572** | **3.973**% | 121.217 | 5.808% |
|  | C=50 | 182.898 | 170.920 | 6.545% | 172.427 | 5.724% | **175.093** | **4.267**% |
| All |  | - |  | 3.552% |  | 2.648% |  | **1.980**% |

Table 2: Experimental results on KP.

|  |  | Greedy | | Beam size 16 | | Beam size 64 | |
|---|---|---|---|---|---|---|---|
| TSP200 | Full graph | **0.79**% | 5s | **0.17**% | 1m | **0.08**% | 5m |
|  | 100KNNs | 1.31% | 3s | 0.23% | 33s | 0.10% | 3m |
| TSP500 | Full graph | 1.71% | 1m | 0.68% | 15m | 0.54% | 1h |
|  | 100KNNs | 2.58% | 18s | 0.92% | 3m | 0.69% | 12m |
|  | 250KNNs | **1.56**% | 32s | **0.67**% | 9m | **0.53**% | 30m |
| TSP1000 | Full graph | **2.34**% | 7m | **1.31**% | 1.8h | **1.19**% | 7.3h |
|  | 100KNNs | 3.34% | 25s | 1.69% | 6m | 1.45% | 24m |
|  | 250KNNs | 2.53% | 1m | 1.43% | 23m | **1.19**% | 1.4h |
| CVRP200 | Full graph | 4.80% | 5s | 2.42% | 1m | 1.82% | 5m |
|  | 100KNNs | 5.18% | 3s | 2.12% | 33s | 1.68% | 3m |
| CVRP500 | Full graph | 4.74% | 1m | 2.10% | 15m | 1.59% | 1h |
|  | 100KNNs | 5.14% | 18s | 2.02% | 3m | 1.74% | 12m |
|  | 250KNNs | **4.58**% | 32s | **1.86**% | 9m | **1.14**% | 30m |
| CVRP1000 | Full graph | 8.00% | 7m | 3.19% | 1.8h | 2.39% | 7.3h |
|  | 100KNNs | 8.25% | 25s | 4.76% | 6m | 3.58% | 24m |
|  | 250KNNs | **7.51**% | 1m | **3.08**% | 23m | **2.28**% | 1.4h |

Table 3: Improving the model performance using a $k$-nearest-neighbor heuristic.

## C    IMPROVING THE MODEL PERFORMANCE WITH A $k$-NEAREST-NEIGHBOR HEURISTIC

Our decoding strategy could be further improved by using a $k$-nearest-neighbor heuristic to restrict the search space and reduce the inference time. For both greedy and beam search strategies, at every step, it is possible to reduce the remaining graph by considering only a certain number of neighbouring nodes. Table 3 presents the experiments on TSP and CVRP where we apply the model just on a certain number on nearest neighbours of the origin. This approach clearly reduces the execution time, but also in some cases even improves the performance in terms of optimality gap. The same heuristic can be applied on Knapsack problem, where model could be applied just on a certain number of items with highest values.

## D    ABLATION STUDY

### D.1    TRANSFORMER VS HYPERMIXER AS MODEL

In Section 6 we have shown that our model has an excellent generalization ability to graphs of larger size. In Section 4, we hypothesize that this has to do with the fact that a subproblem of size $t$ spends $\mathcal{O}(t^2)$ computation operations due to the quadratic complexity of the Transformer encoder's self-attention component, which is responsible for mixing node representations. To test this hypothesis, we experiment with replacing self-attention with an efficient mixing component (see Tay et al. (2022) for an overview), namely the recent linear-time HyperMixer (Mai et al., 2022). We chose this model because it does not assume that the input is ordered, unlike e.g. sparse attention alternatives.

| Seed | TSP100 | TSP200 | TSP500 | TSP1000 |
|------|--------|--------|---------|----------|
| 1 | 2.10% | 8.38% | 34.91% | 71.30% |
| 2 | 1.38% | 3.54% | 98.59% | 628.71% |
| 3 | 1.93% | 4.14% | 120.18% | 216.77% |
| 4 | 1.37% | 4.54% | 46.23% | 104.85% |
| 5 | 1.25% | 3.66% | 61.99% | 524.43% |

Table 4: Experimental results on TSP with HyperMixer for five different seeds.

**Experimental Details** For comparability, we set the model and training parameters to the same as for Transformers, so the experiments only differ in token mixing component that is used. The only other difference is that we used Layer Normalization Ba et al. (2016) instead of ReZero Bachlechner et al. (2021), which leads to more stable training for HyperMixer. Since we observed relatively large sensitivity to model initialization, we are reporting the results for 5 different seeds.

**Results** Table 4 shows the result for HyperMixer with greedy decoding. While the model reaches lower but acceptable performance than Transformers on TSP100, it generalizes poorly to instances of larger size. Moreover, performance is very sensitive to the seed. These results suggest that the computation spent by self-attention is indeed necessary to reach the generalization ability of our model, which increases the compute with the size of the (sub)problem.

## D.2 APPROXIMATED MODEL

As mentioned in Section 5, existing works have also noted the importance of accounting for the change of the state after each action: Xin et al. (2021c;b) claimed that models should recompute the embeddings after each action. However because of the additional training cost, they proposed the following approximation: fixing lower encoder levels and recomputing just the top level with a mask of already visited nodes. They hypothesis a kind of hierarchical feature extraction property that may make the last layers more important for the fine-grained next decision. In contrast, we call our entire model after each construction step; effectively recomputing the embeddings of each state. We hypothesis that this property may explain the superior performance (Table 1) w.r.t MDAM model Xin et al. (2021b). In order to support this hypothesis, we have implemented an approximated version of our model as follows. We fixed the bottom layers of our model and recomputed just the top layer, by masking already visited nodes and adding the updated information (origin and destination tokens for TSP). As expected, inference time is 1.6 times shorter, but performance is severely degraded: we obtained optimality gap of 9.833% (vs 0.540% with original model) on TSP100.

## D.3 REZERO VS BATCHNORM AS NORMALIZATION

Most NCO works that use transformer networks (Kool et al., 2019)(Kwon et al., 2021)(Xin et al., 2021b) use batch normalization(Ioffe & Szegedy, 2015) rather than layer normalization (Ba et al., 2016) in attention layers. We find ReZero normalization (Bachlechner et al., 2021) to work even better. Figure 3 shows the effect of using ReZero compared to batch normalization in our Transformer network. Using it leads to more stable training, better performance, and drastically lower variance between seeds.

## E ON THE IMPACT OF EXPERT SOLUTIONS

Our datasets consist of pairs of a problem instance and a solution (tour). On the other hand, in this paper, we use imitation learning, which requires instead pairs of a problem instance and (expert) trajectory in the MDP. Now, a solution may be obtained from multiple trajectories in the MDP. For example, with TSP, a solution is a loop in a graph, and one has to decide at which node its construction started and in which direction it proceeded. With CVRP, the order in which the subtours are constructed needs also to be decided. Hence, all our datasets are pre-processed to transform solutions into corresponding construction trajectories (a choice for each or even all possible ones). We experimentally observed that this transformation has an impact on the performance. For example, with CVRP, choosing, for each solution, the construction in the order in which LKH3 *displays* it,

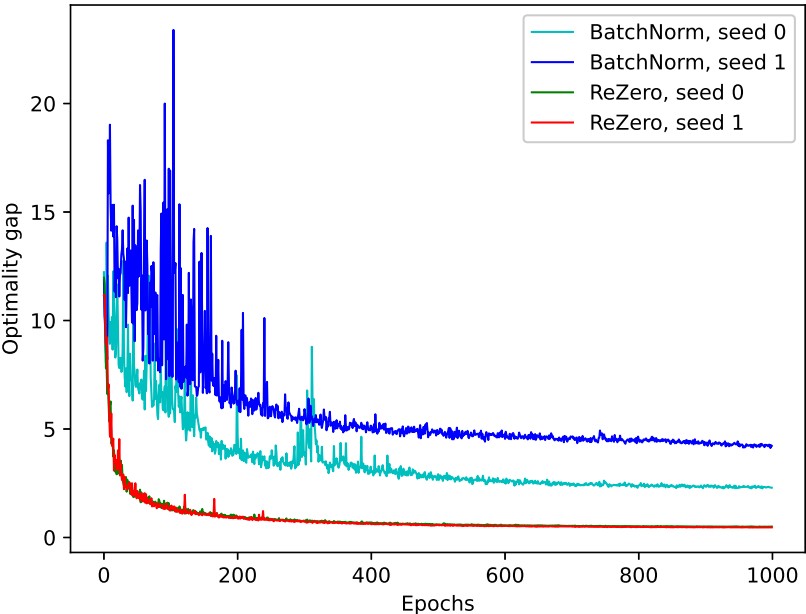

Figure 3: Training curves showing the effect of the choice of normalization layer on validation performance

which does not seem arbitrary, yields to 1.3 point better opt-gap performance compared to following a random ordering of the sub-tours. We hypothesize that if there is any bias in the display of the optimal solution - for example, shorter tour first, or closest node first - it requires slightly less model capacity to learn action imitation *for this display* rather than *for all possible displays*.

## F  PROOF OF PROPOSITION 1 (SOUNDNESS OF THE NAIVE MDP)

We show here that procedure SOLVE satisfies $\text{SOLVE}(f) = \arg\min_{x \in \mathcal{X}} f(x)$. We first show the following general lemma:

Let $\mathcal{Y} \xrightarrow{\psi} \mathcal{X} \xrightarrow{f} \mathbb{R} \cup \{\infty\}$ be arbitrary mappings, if $\psi$ is surjective then

$$\arg\min_{x \in \mathcal{X}} f(x) = \psi(\arg\min_{y \in \mathcal{Y}} f(\psi(y)))$$

Simple application of the definition of $\arg\min$ (as a set). The subscript $_*$ denotes the steps where the assumption that $\psi$ is a surjection is used:

$$
\begin{aligned}
x' \in \psi(\arg\min_y f(\psi(y))) \quad &\text{iff} \quad \exists y' \in \arg\min_y f(\psi(y))\ x' = \psi(y') \\
&\text{iff} \quad \exists y'\ x' = \psi(y')\ \forall y\ f(\psi(y')) \le f(\psi(y)) \\
&\text{iff} \quad \exists y'\ x' = \psi(y')\ \forall y\ f(x') \le f(\psi(y)) \\
&\text{iff}_* \quad \forall y\ f(x') \le f(\psi(y)) \quad \text{iff}_* \quad \forall x\ f(x') \le f(x) \\
&\text{iff} \quad x' \in \arg\min_x f(x)
\end{aligned}
$$

Let $(\mathcal{F}, \mathcal{X})$ be a CO problem with specification $\langle \mathcal{T}, \text{SOL}, \text{VAL} \rangle$ and $\mathcal{M}$ the naive MDP obtained from it. For each $f \in \mathcal{F}$, let $v_f = \text{VAL}(f, \epsilon)$, $\mathcal{X}_f = \{x \in \mathcal{X} | f(x) < \infty\}$ and let $\mathcal{Y}_f$ be the set of $\mathcal{M}$-trajectories which start at $(f, \epsilon)$ and end at a stop state.

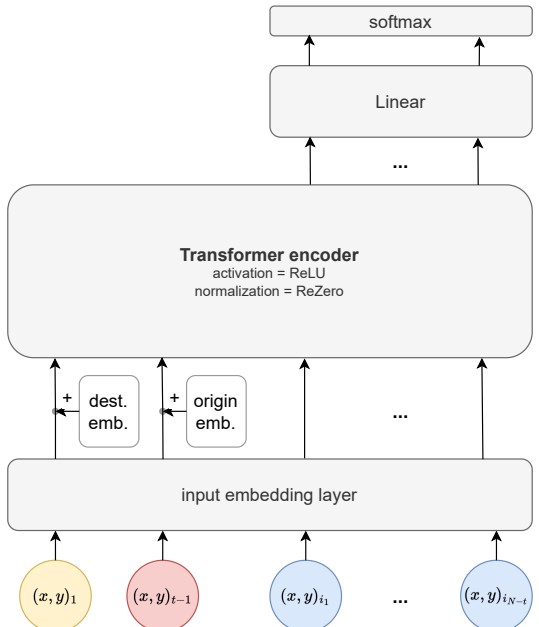

Figure 4: Computation flow at the $t$-th time step, when a partial solution of length $t-1$ already exists. The input state consist of the destination node (i.e. the first and last node in the TSP tour), the origin node (i.e., the most recent node in the tour), and the set of remaining nodes. After passing all input nodes through an embedding layer, we add special, learnable vector embeddings to the origin and current node to signal their special meaning. Finally, a Transformer encoder followed by a linear classifier head selects the next node at step $t$.

- For any $\mathcal{M}$-trajectory $\tau = s_0 t_1 r_1 s_1 \cdots t_n r_n s_n$ in $\mathcal{Y}_f$, define $\psi(\tau) =_{\text{def}} \text{SOL}(s_n)$. Since $\tau \in \mathcal{Y}_f$, we have $s_0 = (f, \epsilon)$ and $s_n$ is a stop state, i.e. $\text{SOL}(s_n) = \psi(\tau) \in \mathcal{X}$, and by Equation 2a, $f(\psi(\tau)) < \infty$. Hence $\psi : \mathcal{Y}_f \mapsto \mathcal{X}_f$.

- By construction, $s_m = (f, t_{1:m})$ for all $m \in 1{:}n$ and each transition in $\tau$ has a finite reward $\text{VAL}(s_{m-1}) - \text{VAL}(s_m)$ (condition for it to be valid). Hence the cumulated reward is given by $R(\tau) = \text{VAL}(s_0) - \text{VAL}(s_n)$. Now, $\text{VAL}(s_0) = v_f$ which is independent of $\tau$ and by Equation 2c, $\text{VAL}(s_n) = f(\psi(\tau))$. Hence $f(\psi(\tau)) = v_f - R(\tau)$.

- Let's show that $\psi$ is surjective. Let $x \in \mathcal{X}_f$. Equation 2a ensures that $x = \text{SOL}(f, t_{1:n})$ for some $t_{1:n} \in \mathcal{T}^*$. For each $m \in \{0{:}n\}$, let $s_m = (f, t_{1:m})$ and consider the sequence $\tau = s_0 t_1 r_1 s_1 \cdots t_n r_n s_n$. Now, $\text{SOL}(s_n) = x \neq \bot$ hence $\tau$ ends in a stop state and starts at $(f, \epsilon)$. By Equation 2c we have $\text{VAL}(s_n) = f(x)$, hence $\text{VAL}(s_n) < \infty$, and $\text{VAL}(s_m) < \infty$ for all $m \in \{0{:}n-1\}$. And by Equation 2b $\text{SOL}(s_m) = \bot$, hence all the transitions in $\tau$ are valid in $\mathcal{M}$. Hence $\tau \in \mathcal{Y}_f$ and by definition, $\psi(\tau) = x$.

Therefore we can apply the lemma proved above:

$$\arg \min_{x \in \mathcal{X}_f} f(x) = \psi(\arg \min_{\tau \in \mathcal{Y}_f} f(\psi(\tau))) = \psi(\arg \min_{\tau \in \mathcal{Y}_f} v_f - R(\tau))$$

$$= \psi(\arg \max_{\tau \in \mathcal{Y}_f} R(\tau)) = \psi(\text{SOLVE}_{\mathcal{M}}^{\text{MDP}}(f, \epsilon)) = \text{SOLVE}(f)$$

Now, obviously, $\arg \min_{x \in \mathcal{X}} f(x) = \arg \min_{x \in \mathcal{X}_f} f(x)$, since by definition $f$ is infinite on $\mathcal{X} \backslash \mathcal{X}_f$.

# G    PLOTS OF SOME TSPLIB AND CVRPLIB SOLUTIONS

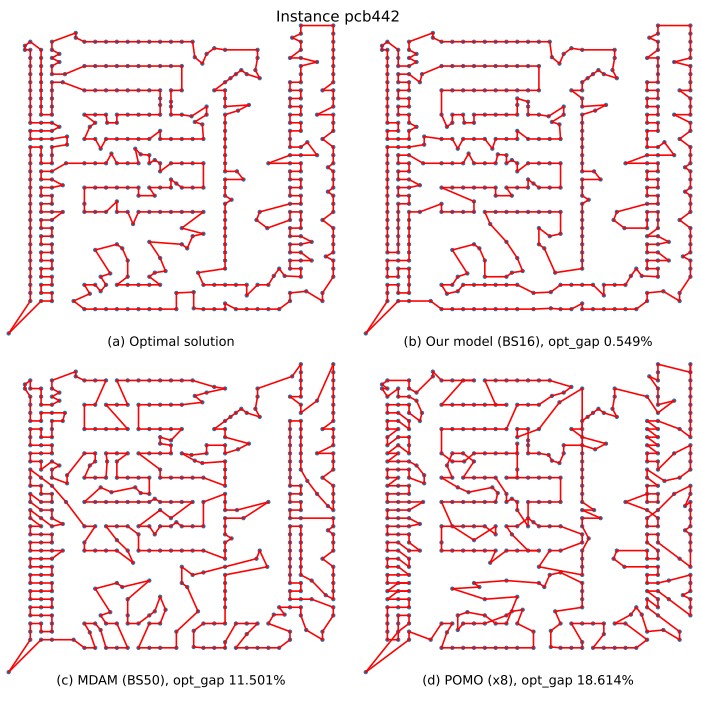

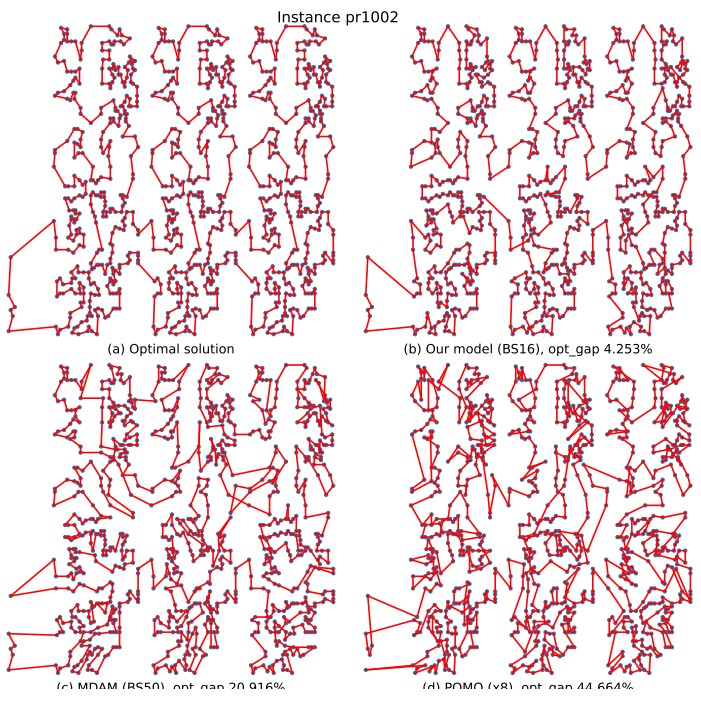

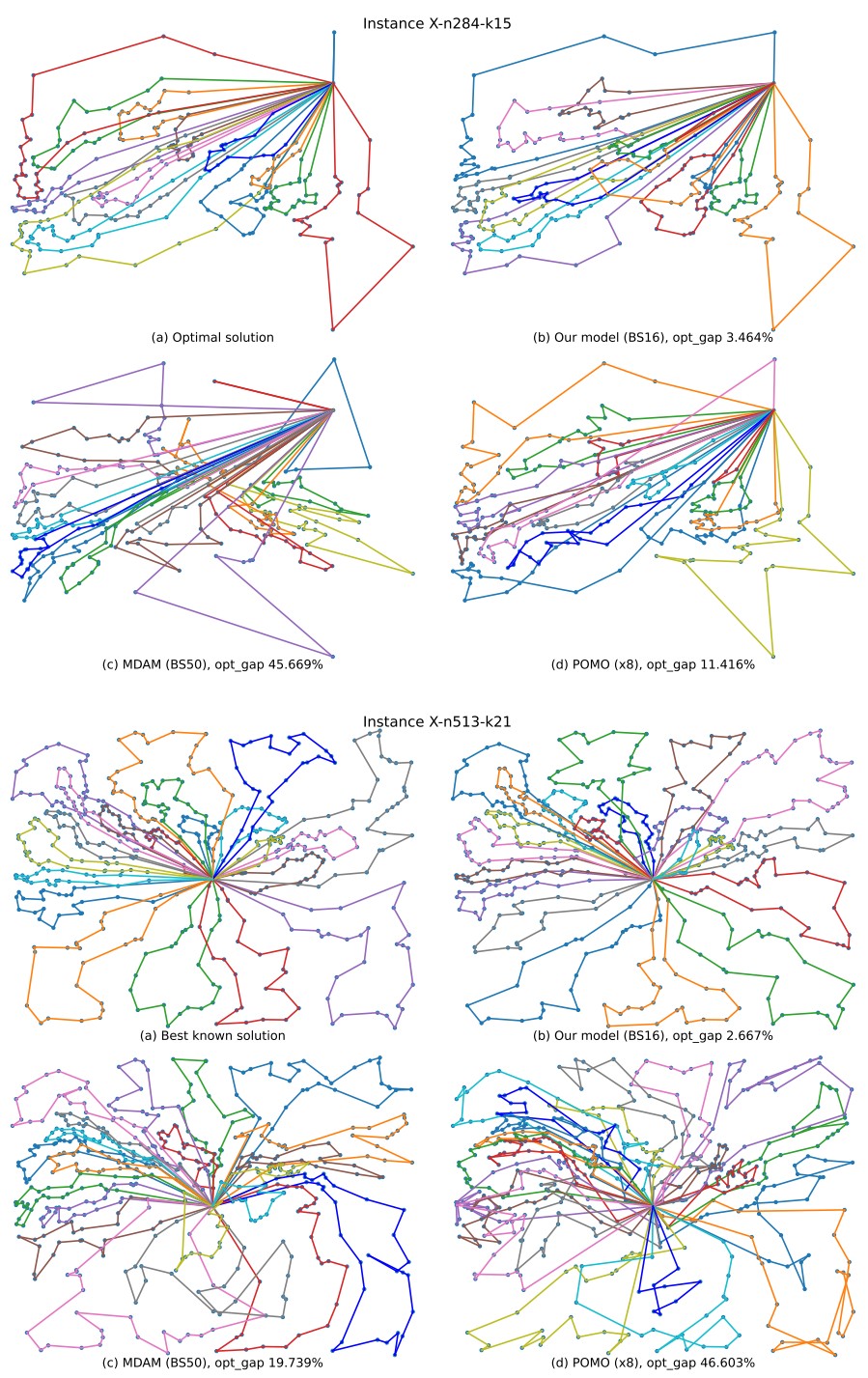

## H  BACKGROUND ON BISIMULATION-BISIMILARITY

### H.1  BISIMULATION IN LABELLED TRANSITION SYSTEMS

Bisimulation is a very broad concept which applies to arbitrary Labelled Transition Systems (LTS). It has been declined in various flavours of LTS, such as Process Calculi, Finite State Automata, Game theory, and of course MDP (initially deterministic MDP such as those used here, later extended to stochastic MDP which we are not concerned with here). A bisimulation is a binary relation $\mathcal{R}$ among states which "commutes" with the transitions of the LTS in the following diagram, which should informally be read as follows: if the pair of arrows connected to $p$ (resp. $q$) exists then so does the "opposite" pair (w.r.t. the centre of the diagram).

$$
\begin{array}{ccc}
p & \xrightarrow{\ell} & p' \\
\mathcal{R}\downarrow & \xrightarrow{\ell} & \downarrow\mathcal{R} \\
q & \xrightarrow{\ell} & q'
\end{array}
$$

The notation $p \xrightarrow{\ell} p'$ means the transition from $p$ to $p'$ with label $\ell$ is valid. Thus, formally,

**Definition 1.** *A binary relation $\mathcal{R}$ on states is a bisimulation if for all label $\ell$ and states $p, q$ such that $p\mathcal{R}q$*

$$\forall p' \text{ s.t. } p \xrightarrow{\ell} p' \;\exists q' \text{ s.t. } q \xrightarrow{\ell} q' , \; p'\mathcal{R}q' \qquad\qquad \forall q' \text{ s.t. } q \xrightarrow{\ell} q' \;\exists p' \text{ s.t. } p \xrightarrow{\ell} p' , \; p'\mathcal{R}q'$$

Note that this definition is extended to the "heterogeneous" case where $\mathcal{R}$ is bi-partite, relating the state spaces of two LTS $\mathcal{L}_1, \mathcal{L}_2$ *sharing the same label space*. One just forms a new LTS $\mathcal{L}$ whose state space is the *disjoint* union of the state spaces of $\mathcal{L}_1, \mathcal{L}_2$ and the transitions are those of $\mathcal{L}_1, \mathcal{L}_2$ in their respective (disjoint) component. An heterogeneous bisimulation on $\mathcal{L}_1, \mathcal{L}_2$ is then a (homogeneous) bisimulation on $\mathcal{L}$. Most results below also have a heterogeneous version.

**Proposition 2.** *The set of bisimulations (subset of the set of binary relations on states) is stable by union, composition, and inversion, hence also by reflexive-symmetric-transitive closure.*

In particular, the union of all bisimulations, called the *bisimilarity* of the LTS, is itself a bisimulation, and it is also an equivalence relation.

*Proof.* (outline) Let's detail stability by composition, the other cases are similarly obvious. If $\mathcal{R}_1, \mathcal{R}_2$ are the two bisimulations being composed, apply the commutation property to each cell of the following diagram (from top to bottom).

$$
\begin{array}{ccc}
p & \xrightarrow{\ell} & p' \\
\mathcal{R}_1\downarrow & \xrightarrow{\ell} & \downarrow\mathcal{R}_1 \\
r & \xrightarrow{\ell} & r' \\
\mathcal{R}_2\downarrow & \xrightarrow{\ell} & \downarrow\mathcal{R}_2 \\
q & \xrightarrow{\ell} & q'
\end{array}
$$

$\square$

**Definition 2.** *Given an LTS $\mathcal{L}$, its* transitive closure *is another LTS denoted $\mathcal{L}^*$ on the same state space, where the labels are the sequences of labels of $\mathcal{L}$ and the transitions are defined by*

$$p \xrightarrow[(\mathcal{L}^*)]{\ell_{1:n}} p' \quad \text{if} \quad \exists p_{0:n} \text{ such that } p = p_0 \xrightarrow[(\mathcal{L})]{\ell_1} p_1 \cdots \xrightarrow[(\mathcal{L})]{\ell_{n-1}} p_{n-1} \xrightarrow[(\mathcal{L})]{\ell_n} p_n = p'$$

**Proposition 3.** *If $\mathcal{R}$ is a bisimulation on $\mathcal{L}$, then it is also a bisimulation on $\mathcal{L}^*$.*

*Proof.* (outline) This is essentially shown by successively applying the commutation property to each cell of the following diagram (from left to right):

$$p_0 \xrightarrow{\ell_1} p_1 \dashrightarrow p_{n-1} \xrightarrow{\ell_n} p_n$$
$$\mathcal{R}\!\downarrow \quad \ell_1 \quad \mathcal{R}\!\downarrow \qquad\qquad \downarrow\mathcal{R} \quad \ell_n \quad \downarrow\mathcal{R}$$
$$q_0 \xrightarrow{\phantom{\ell_1}} q_1 \dashrightarrow q_{n-1} \xrightarrow{\phantom{\ell_n}} q_n$$

$\square$

**Definition 3.** *Given an LTS $\mathcal{L}$ and an equivalence relation $\mathcal{R}$ on its state space, we can define the quotient LTS $\mathcal{L}/\mathcal{R}$ with the same label space, where the states are the $\mathcal{R}$-equivalence classes and the transitions are defined, for any classes $\dot{p}, \dot{p}'$, by*

$$\dot{p} \xrightarrow[\mathcal{L}/\mathcal{R}]{\ell} \dot{p}' \quad if \quad \forall p \in \dot{p} \; \exists p' \in \dot{p}' \quad p \xrightarrow[\mathcal{L}]{\ell} p'$$

**Proposition 4.** *Let $\mathcal{R}$ be an equivalence on the state space of $\mathcal{L}$. $\mathcal{R}$ is a bisimulation on $\mathcal{L}$ if and only if $\in$ is a (heterogeneous) bisimulation on $\mathcal{L}, \mathcal{L}/\mathcal{R}$.*

*Proof.* We show both implications:

- Assume $\mathcal{R}$ is a bisimulation on $\mathcal{L}$.

    - Let $p \in \dot{q}$ and $p \xrightarrow{\ell} p'$. Let $q \in \dot{q}$. Hence $p\mathcal{R}q$ and $p \xrightarrow{\ell} p'$. Since $\mathcal{R}$ is a bisimulation, there exists $q'$ such that $q \xrightarrow{\ell} q'$ and $p'\mathcal{R}q'$. Hence for all $q \in \dot{q}$, there exists $q' \in \bar{p}'$ such that $q \xrightarrow{\ell} q'$. Hence by definition $\dot{q} \xrightarrow{\ell} \bar{p}'$ while $p' \in \bar{p}'$.
    - Let $p \in \dot{q}$ and $\dot{q} \xrightarrow{\ell} \dot{q}'$. Hence by definition, there exists $p' \in \dot{q}'$ such that $p \xrightarrow{\ell} p'$.

- Assume $\in$ is a (heterogeneous) bisimulation on $\mathcal{L}, \mathcal{L}/\mathcal{R}$.

    - Let $p\mathcal{R}q$ and $p \xrightarrow{\ell} p'$. Hence $p \in \bar{q}$ and $p \xrightarrow{\ell} p'$. Since $\in$ is a bisimulation, there exists $\dot{q}'$ such that $p' \in \dot{q}'$ and $\bar{q} \xrightarrow{\ell} \dot{q}'$. Now $q \in \bar{q}$, hence, by definition, there exists $q' \in \dot{q}'$ such that $q \xrightarrow{\ell} q'$. And $p'\mathcal{R}q'$ since $p', q' \in \dot{q}'$.
    - Let $p\mathcal{R}q$ and $q \xrightarrow{\ell} q'$. Hence $q\mathcal{R}p$ and $q \xrightarrow{\ell} q'$, and we are in the previous case up to a permutation of variables. $\square$

**Proposition 5.** *Let $\mathcal{R}$ be an equivalence relation on the state space of $\mathcal{L}$. If $\mathcal{R}$ is a bisimulation on $\mathcal{L}$, then for any $\mathcal{L}$-state $p$, $\mathcal{L}/\mathcal{R}$-state $\dot{p}$ and $\mathcal{L}^*$-label $\ell$*

$$\bar{p} \xrightarrow[(\mathcal{L}/\mathcal{R})^*]{\ell} \dot{p}' \quad if \text{ and only if} \quad \exists p' \in \dot{p}' \quad p \xrightarrow[\mathcal{L}^*]{\ell} p'$$

*Proof.* Simple combination of Propositions 4 and 3. $\mathcal{R}$ is a bisimulation on $\mathcal{L}$, hence $\in$ is a heterogeneous bisimulation on $\mathcal{L}, \mathcal{L}/\mathcal{R}$ (Proposition 4), hence also a heterogeneous bisimulation on $\mathcal{L}^*, (\mathcal{L}/\mathcal{R})^*$ (Proposition 3, heterogeneous version).

- If $\bar{p} \xrightarrow[(\mathcal{L}/\mathcal{R})^*]{\ell} \dot{p}'$, since $p \in \bar{p}$ and $\in$ is a bisimulation, we have $p \xrightarrow[\mathcal{L}^*]{\ell} p'$ for some $p' \in \dot{p}'$.

- Conversely, if $p \xrightarrow[\mathcal{L}^*]{\ell} p'$ for some $p' \in \dot{p}'$, since $p \in \bar{p}$ and $\in$ is a bisimulation, we have $\bar{p} \xrightarrow[(\mathcal{L}/\mathcal{R})^*]{\ell} \dot{q}'$ and $p' \in \dot{q}'$ for some $\dot{q}'$. Now $p' \in \dot{p}' \cap \dot{q}'$ hence $\dot{p}' = \dot{q}'$ and $\bar{p} \xrightarrow[(\mathcal{L}/\mathcal{R})^*]{\ell} \dot{p}'$. $\square$

### H.2 BISIMULATION IN DETERMINISTIC MDP

**Definition 4.** *An MDP is a pair $(\mathcal{L}, \top)$ where $\mathcal{L}$ is a LTS with label space $\mathcal{A} \times \mathbb{R}$ for some action space $\mathcal{A}$ (action-reward pairs denoted $a|r$) and $\top$ is a subset of states (the* stop *states). It is said to be* deterministic *if*

$$if \; s \xrightarrow{a|r_1} s_1' \text{ and } s \xrightarrow{a|r_2} s_2' \text{ then } r_1 = r_2 \text{ and } s_1' = s_2'$$

*Given an $\mathcal{L}$-trajectory $\tau$, i.e. a sequence $s_0 a_1 r_1 s_1 \cdots a_n r_n s_n$ where $s_{i-1} \xrightarrow{a_i|r_i} s_i$ for all $i \in \{1{:}n\}$, its* cumulated reward *is defined by $R(\tau) = \sum_{i=1}^n r_i$. The generic problem statement of the MDP solution framework is, given an MDP $(\mathcal{L}, \top)$ and one of its states $s_o$, to solve the following optimisation:*

$$\text{SOLVE}^{MDP}((\mathcal{L}, \top), s_o) = \arg \max_{\tau} R(\tau) \mid \tau \text{ is a } \mathcal{L}\text{-trajectory starting at } s_o \text{ and ending in } \top$$

This definition of MDP and the standard textbook one coincide only in the deterministic case (in the standard definition, an MDP is deterministic if the distribution of output state-reward pairs for a given input state and allowed action is "one-hot"). The non deterministic case in the definition above does not match the standard definition: it would be wrong to interpret two distinct transitions for the same input state $s$ and action $a$ as meaning that the outcome of applying $a$ to state $s$ is distributed between the two output reward-state pairs according to a specific distribution (e.g. uniform). Also, in the problem statement, the objective $R(\tau)$ has no expectation, which, with the standard definition, only makes sense in the case of a deterministic MDP. Similarly, the standard problem statement is expressed in terms of policies rather than trajectories directly, but in the deterministic case, the two are equivalent. Observe that there is a one-to-one correspondence between trajectories in $\mathcal{L}$ and transitions in the LTS $\mathcal{L}^*$, so the problem statement can be formulated equivalently as

$$\text{SOLVE}^{MDP}((\mathcal{L}, \top), s_o) = \arg \max_{\ell} R(\ell) \mid \exists s \in \top, s_o \xrightarrow[\mathcal{L}^*]{\ell} s \tag{4}$$

**Proposition 6.** *Let $(\mathcal{L}, \top)$ be an MDP and $\mathcal{R}$ an equivalence relation on its state space.*

1. *$(\mathcal{L}/\mathcal{R}, \bar{\top})$ is also an MDP, where $\bar{\top} = \{\bar{s} | s \in \top\}$, and if $\mathcal{L}$ is deterministic, so is $\mathcal{L}/\mathcal{R}$.*

2. *If $\mathcal{R}$ is a bisimulation on $\mathcal{L}$ preserving $\top$ (i.e. $\bigcup_{s \in \top} \bar{s} = \top$), then for any state $s_o$ and label $\ell$ in $\mathcal{L}^*$ we have*

$$\exists s \in \top, s_o \xrightarrow[\mathcal{L}^*]{\ell} s \quad \text{if and only if} \quad \exists \dot{s} \in \bar{\top}, \bar{s}_o \xrightarrow[(\mathcal{L}/\mathcal{R})^*]{\ell} \dot{s}$$

*Proof.* The second property is a direct consequence of Proposition 5 and the assumption that $\top$ is preserved by $\mathcal{R}$. For the first, assume that $\mathcal{L}$ is deterministic. Let $\dot{s}, \dot{s}_1, \dot{s}_2$ be $\mathcal{L}/\mathcal{R}$ states, such that $\dot{s} \xrightarrow{a|r_1} \dot{s}_1$ and $\dot{s} \xrightarrow{a|r_2} \dot{s}_2$. Choose $s \in \dot{s}$. Hence, by definition, there exist $s_1 \in \dot{s}_1$ and $s_2 \in \dot{s}_2$ such that $s \xrightarrow{a|r_1} s_1$ and $s \xrightarrow{a|r_2} s_2$. Since $\mathcal{L}$ is deterministic, we have $r_1 = r_2$ and $s_1 = s_2 \in \dot{s}_1 \cap \dot{s}_2$, hence $\dot{s}_1 = \dot{s}_2$. Hence $\mathcal{L}/\mathcal{R}$ is also deterministic. $\square$

Therefore, when $\mathcal{R}$ is a bisimulation equivalence on $\mathcal{L}$ preserving $\top$, the generic MDP problem statement of Eq. equation 4 can be reformulated as

$$\text{SOLVE}^{MDP}((\mathcal{L}, \top), s_o) = \text{SOLVE}^{MDP}((\mathcal{L}/\mathcal{R}, \bar{\top}), \bar{s}_o) = \arg \max_{\ell} R(\ell) \mid \exists \dot{s} \in \bar{\top}, \bar{s}_o \xrightarrow[(\mathcal{L}/\mathcal{R})^*]{\ell} \dot{s} \tag{5}$$

Note that a bisimulation on $\mathcal{L}$ preserving $\top$ is simply a bisimulation on the LTS $\dot{\mathcal{L}}$ defined as follows: $\dot{\mathcal{L}}$ has the same state space as $\mathcal{L}$ and an additional transition $s \xrightarrow{\cdot} s$ for each $s \in \top$, where "$\cdot$" is a distinguished label not present in $\mathcal{L}$.

A bisimulation $\mathcal{R}$ on $\dot{\mathcal{L}}$ captures some symmetries of the state space of $\dot{\mathcal{L}}$. If $\mathcal{R}$ is taken to be the bisimilarity of $\dot{\mathcal{L}}$, i.e. the union of all the bisimulations on $\dot{\mathcal{L}}$, i.e. the union of all the bisimulations on $\mathcal{L}$ preserving $\top$, then it captures all the possible symmetries of the state space. This should be seen as an asymptotic result, since constructing and working with the full bisimilarity of $\dot{\mathcal{L}}$ is not feasible. But Proposition 6 remains valuable as it applies to all bisimulation, not just the maximal bisimulation of $\dot{\mathcal{L}}$ (its bisimilarity).

