# OpenReview forum: "BQ-NCO: Bisimulation Quotienting for Generalizable Neural Combinatorial Optimization"
_ICLR.cc/2023/Conference — Submitted to ICLR 2023_

### Official Review · Reviewer_yAnE · 2022-10-22

**Confidence:** 4
**Correctness:** 3
**Technical Novelty And Significance:** 2
**Empirical Novelty And Significance:** 3
**Recommendation:** 3

**Clarity, Quality, Novelty And Reproducibility:**

The clarity could be improved by providing more formal definitions of the involved terms.

The quality and novelty are less satisfactory due to the lack of details.

The reproducibility could be better supported by providing course codes.


**Strength And Weaknesses:**

Strength:

The overall idea is reasonable: capturing symmetries in the system is expected to reduce the required training efforts and thereby facilitate good generalization. Most of the sections are well-written (module the technical part), and the studied problem is well-motivated. The presented experimental results suggest that the proposed method is promising.


Weaknesses:

- W1: The presentation of this paper lacks mathematical rigor, which makes its technical contribution less clear.
  - Is $T$ a set of steps or a set of sequences of steps? Is $T^*$ a closure of $T$? Some notations on pages 2 and 3 are inconsistent.
  - It would be better to formally define the concepts of being finalized, the new space $\hat{S}$.
  - Since the process is deterministic, should there be only one expansion?
  - I am a little confused by the context around Proposition 1, especially by the claim that MDP is equivalent to solving the CO problem. The proof seems to show that MDP can simulate general Turing machines subsuming CO algorithms, which is well-known.
  - The mapping $\Theta$ introduced in Section 3 is desired but computationally hard to obtain in general. In addition, its role in the proposed model (Sec 4) is not very clear.

- W2: Another major concern is that the designed model, as the main contribution, was described in a very abstract manner, and it is not clear how the ideas in the previous sections can be realized. Therefore, the novelty of this paper is not clear. The concrete contribution of this paper is entirely in the first paragraph of Section 4, but there are almost no technical details. As such, it seems that the proposed method can be obtained from the existing methods with very slight modifications.

- W3: The paper could support reproducibility by including the source code.


**Summary Of The Paper:**

This paper studies the problem of solving combinatorial optimization problems using reinforcement learning enhanced with bisimulation quotients, where the main idea is to reduce the search space by identifying states with identical paths in MDP. In particular, new models are designed for TSP and CVRP. The proposed methods are supplemented with experimental studies


**Summary Of The Review:**

The paper presents an interesting idea but could do better in explaining its novelty by having more details. The overall contribution seems limited.

---

> ### Author Response · Authors · 2022-11-18
> **Reply to reviewer yAnE**
>
> Thank you for the feedback. We address below your different comments and we’ll be happy to provide more details to clarify any further points.
>
> **W1. "Is $T$ a set of steps or a set of sequences of steps? Is $T^{\star}$ a closure of $T$?"** As written in the paper $\mathcal{T}$ is a set of steps. $\mathcal{T}^{\star}$ is the set of sequences of elements of  $\mathcal{T}.$ We have clarified this point in the manuscript.
>
> **W1. "Some notations on pages 2 and 3 are inconsistent."** We have again carefully reviewed our notations. Could you please point out the inconsistencies?
>
> **W1. "It would be better to formally define the concepts of being finalized, the new space $\hat{S}$."** A partial solution $s$ is finalized if $\text{SOL}(s)$ is a feasible solution. We have clarified this in the paper. We do not understand what is meant by the new space $\hat{S}$, could you please elaborate?
>
> **W1. "Since the process is deterministic, should there be only one expansion?"** The construction MDP is deterministic in the sense that given a state (e.g. a partial solution) and action (a construction step), the next state is always the same. This is due to the deterministic nature of the CO problems that we address. Since at each step, there are generally many possible actions, there is generally an exponential number of possible expansions.
>
> **W1. "I am a little confused by the context around Proposition 1, especially by the claim that MDP is equivalent to solving the CO problem. The proof seems to show that MDP can simulate general Turing machines subsuming CO algorithms, which is well-known."** Given a CO problem and its specification $\langle\tau,SOL,VAL\rangle$, we present a way to automatically derive a naive MDP to construct solutions.  Proposition 1 states the equivalence between solving this naive MDP and solving the original CO problem.
>
> **W1. "The mapping $\Theta$ introduced in Section 3 is desired but computationally hard to obtain in general. In addition, its role in the proposed model (Sec 4) is not very clear."** We assume the reviewer is referring to the mapping $\Phi$, i.e. the bisimulation. We agree that the ideal **bisimilarity** mapping is indeed generally intractable. However our results hold for **bisimulations** (see Section 3 - “Bisimulation quotienting”) and we argue that such bisimulations can be naturally defined for many CO problems by exploiting their symmetries. Intuitively, the question is, after a number of construction steps, what is the minimal information that is needed to choose the next steps. In addition to the 2 examples of bisimulations for TSP and CVRP presented in the paper, we have provided bisimulation examples for two other classical non-routing CO problems: the knapsack problem and the maximum vertex cover problem – see General Reply above.
>
> **W2. "Another major concern is that the designed model, as the main contribution, was described in a very abstract manner, and it is not clear how the ideas in the previous sections can be realized. Therefore, the novelty of this paper is not clear. The concrete contribution of this paper is entirely in the first paragraph of Section 4, but there are almost no technical details. As such, it seems that the proposed method can be obtained from the existing methods with very slight modifications."** Indeed, the main component of our neural architecture is a Transformer encoder with only small modifications from the original formulation. Since the general audience at ICLR is very familiar with the Transformer model, we didn’t reiterate it in technical detail. For easier understanding, we added a visualization of the model to the Appendix as Figure 4. The novelty of our method lies in **how** the neural architecture is applied rather than the architecture itself: Its input is the BQ-MDP state (the origin node, destination node, and all remaining nodes, NO partial solution) rather than the naive MDP state that previous papers are based on. As you correctly point out, these ideas were derived theoretically in Sections 2 and 3, and constitute a major part of our contribution. So it is true that our model can be obtained easily from existing methods with small modifications. Yet, despite few simple changes, which are grounded in theory, our model achieves outstanding state-of-the-art generalization performance. We consider this a strength rather than a weakness.
>
> **W3. "The paper could support reproducibility by including the source code."** We are committed to the open science movement and agree that publishing the source code will increase the reproducibility of our paper. For this reason we committed to publishing the code upon acceptance. Note, however, that many organizations don’t allow publishing **anonymized** code because of licensing and attribution issues. Meanwhile, we took several measures to facilitate the reproducibility of our results – see Reproducibility Statement.

---

### Official Review · Reviewer_Sx6V · 2022-10-25

**Confidence:** 3
**Correctness:** 3
**Technical Novelty And Significance:** 2
**Empirical Novelty And Significance:** 3
**Recommendation:** 5

**Clarity, Quality, Novelty And Reproducibility:**

- The paper is mostly clear, however as I noted before the theoretical part seems somewhat disconnected from the actual neural approach used.

- The approach provide a novel mechanism for exploiting symmetry in neural combinatorial optimization

- Reproducibility: the paper and the appendix seem to provide sufficient details for reproducing the results. The code is not attached however authors state they plan to make the code public upon acceptance.


**Strength And Weaknesses:**

Strengths:
* Novel and simple approach to enhance neural combinatorial optimization problem by exploiting symmetry
* Results on generalization to larger instances than the training set instances show impressive improvement over the baselines


Weaknesses:
* While the approach is presented as a generic and flexible framework for combinatorial optimization, it seems to require a domain-specific mapping (Phi). The paper only considers such mapping for two routing problems and it is not clear if such mappings would naturally exist for many other (particularly non-routing) combinatorial problems and whether they will prove useful in such problems.
* The long theoretical discussion on general combinatorial optimization using MDP and bisimulation-quotienting seems somewhat disconnected from the actual neural approach. While it does seem to inspire such approach, the proposed neural approach could stand on its own as a new way of handling symmetries in routing problems. In particular, unlike the MDP framework built around rewards, the neural approach is trained using imitation.
* The approach requires optimal solutions for training. Such optimal solutions are typically intractable to compute beyond certain problem size. This is an important difference from approaches like Kool et al. (2021) based on reinforcement learning that do not rely on ground-truth optimal solutions.

Minor question: does "AM bs1024" indicate a beam search with beam width of 1024?

**Summary Of The Paper:**

The paper presents a framework for defining construction heuristic MDPs for combinatorial optimization problems and proposes a technique for exploiting symmetries in state representation in such MDPs based on a mapping to a different state representation. They design a transformer-based architecture for solving TSP and CVRP that utilizes such mapping and show that they significantly improve over the baseline in terms of generalization to larger problems.

**Summary Of The Review:**

The paper seems to propose a nice enhancement to neural combinatorial optimization by exploiting symmetries in states. The experimental results show impressive improvement over baselines on problems of larger size compared to training set. The main weaknesses are: (1) it is not entirely clear how much the experimental results benefit from the theory presented in the paper; (2) the existing experimental results are limited to two routing problems and it is not clear that this technique would easily extend to non-routing combinatorial problems despite being presented as a general approach for combinatorial problems.

---

> ### Author Response · Authors · 2022-11-18
> **Reply to Reviewer Sx6V**
>
> Thank you for your review. We provide answers to each of your concerns below.
>
> **"While the approach is presented as a generic and flexible framework for combinatorial optimization, it seems to require a domain-specific mapping (Phi)."** We confirm that the bisimulation $\Phi$ is problem-specific; it should exploit the problem’s specific symmetries. We claim our framework is generic and flexible because given any problem specification, we can automatically derive a naive MDP for a constructive heuristic (Sec 2-“Solution construction as an MDP” paragraph). Then given any problem-specific bisimulation $\Phi$, we can automatically derive the BQ-MDP induced by $\Phi$ (following Sec 3 - “Bisimulation Quotienting” paragraph).
>
> **"The paper only considers such mapping for two routing problems and it is not clear if such mappings would naturally exist for many other (particularly non-routing) combinatorial problems and whether they will prove useful in such problems."** We argue that such mapping should be easy to define, even beyond routing problems. In the general reply, we have defined $\Phi$ for the knapsack problem and the Minimum Vertex Cover Problem. Experimentally, we find that the BQ transformation for the KP manages to outperform the strong POMO baseline. This shows that the benefit of our model is not limited to routing problems only.
>
> **"The long theoretical discussion on general combinatorial optimization using MDP and bisimulation-quotienting seems somewhat disconnected from the actual neural approach. While it does seem to inspire such approach, the proposed neural approach could stand on its own as a new way of handling symmetries in routing problems. In particular, unlike the MDP framework built around rewards, the neural approach is trained using imitation."**  Our neural method is special in **how** the neural architecture is applied rather than the architecture itself: Its input is the BQ-MDP state (the origin node, destination node, and all remaining nodes, NO partial solution) rather than the naive MDP state that some previous works are based on. These ideas were derived theoretically in Sections 2 and 3. While Section 4 could in principle stand on its own, Section 2 and 3 provide the theoretical foundations and thus constitute a major part of our contribution.
> Note that our method transforms the MDP itself and is not tied to a specific way of solving the MDP (or training the neural model). For the CO problems considered in this paper, we found imitation learning from relatively small instances (100 nodes), which are feasible to solve optimally, sufficient, because the generalization ability of our model to larger instances is excellent. While training our models via reinforcement learning is interesting, it is not a variable of interest in this paper, so we consider it out of scope.
>
> **"The approach requires optimal solutions for training. Such optimal solutions are typically intractable to compute beyond certain problem size. This is an important difference from approaches like Kool et al. (2021) based on reinforcement learning that do not rely on ground-truth optimal solutions."**
> We would like to emphasize that the proposed framework and BQ-reduction for constructive CO heuristics does *not* require supervised training. Transforming the MDP is done independently of how the MDP is solved later on. The BQ-MDP **can** also be solved with reinforcement learning, for example. Computing optimal solutions for NP-hard CO problems is intractable beyond a certain size, while relatively easy for small sizes. Therefore we believe that a supervised approach only makes sense if the training is done on relatively small instances. But of course, for the trained neural model to be useful, it would need to either (i) be much faster than what was used to generate the ground-truth; or (ii) solve instances that are more challenging than the training ones – typically of larger size. This is exactly what we provide by training on instances that are easy to solve for specialized solvers (N=100 for TSP/CVRP) while having an excellent generalization performance on large instances (up to N=1000). Note that this is a key difference with [Kool et al 2019] and similar works, where training and test is done on the same size; and the generalization performance on larger sizes drastically degrades (see Table 1).
>
> **"does AM bs1024 indicate a beam search with beam width of 1024?"** Yes. Although the original work only reports results for greedy rollouts and sampling, we use beam search (which is implemented in their original code repository) because it gave the best results.

---

> > ### Comment · Reviewer_Sx6V · 2022-12-11
> > **Thank you**
> >
> > Thank you for your response.
> >
> > The additional experiments on the knapsack problem are an important addition, however the improvement in performance seems weaker compared to the results presented in the paper.
> >
> > Unfortunately, the response does not fully address my concerns regarding the problem-specific nature of the method, the weak connection between the theoretical part and the neural approach, or the dependence on optimal solutions in the results presented in the paper.

---

> > > ### Author Response · Authors · 2022-12-12
> > > **Connection between our theoretical framework and neural model and genericity of the approach**
> > >
> > > Thank you for the feedback. We would like to emphasis a couple of points:
> > >
> > >  * Regrading the connection between the proposed theoretical framework and the neural model, as previously mentioned, the theory essentially gives the BQ-MDPs and the proposed neural model is a simple Transformer (like many previous works) but *adapted to the BQ-reduced state input*. We have provided a **new ablation*** that compares our model to a slightly modified version, that takes as input the naive MDP state. The significant drop in performance confirms the key role of using the theoretically-grounded BQ-reduced state in the obtained results.
> > >
> > >  * Regarding the genericity/problem-specific nature of our method, we would like to emphasis that in addition to the **TSP, CVRP, KP** and **Maximum Vertex Cover**, we have shown in detail** how our method could be applied to solve **Mixed Integer Linear Programs**, which encompass a wide range of combinatorial problems.
> > >
> > >  * While the application of our framework to 3 CO problems led to excellent results, with new state-of-the-art performance on large TSP/CVRP instances, further implementation and experimentation for other problems constitutes promising future work.
> > >
> > >
> > > \* See comment “New ablation study on the use of the BQ-MDP state”
> > > ** See comment “Applicability of the proposed framework to Mixed Integer Programs”

---

### Official Review · Reviewer_kF44 · 2022-10-25

**Confidence:** 4
**Correctness:** 3
**Technical Novelty And Significance:** 3
**Empirical Novelty And Significance:** 3
**Recommendation:** 5

**Clarity, Quality, Novelty And Reproducibility:**

Clarity and Reproducibility: Some description of the method is unclear.

Quality and Novelty: The idea of reducing the state space based on BQ is interesting.


**Strength And Weaknesses:**

Strengths:
* Introducing bisimulation quotienting to reduce the state space of CO problems is an interesting idea.
* The proposed method significantly improves the generalization performance in solving the TSP and CVRP, especially on the problems with large-scale graphs.

Weaknesses:
* The authors claim that their formulation is generic for arbitrary CO problems. However, the formulation seems specific for the TSP and CVRP. Thus, it would be more convincing if the authors could explain how the proposed formulation applies to other CO problems, such as mixed-integer linear programs.
* The details about introducing the bisimulation quotienting into their proposed transformer-based model are unclear. For example, please illustrate the proposed model architecture and hyperparameters. Moreover, how does the proposed method learn two special embedding vectors for the origin and destination nodes?
* The ablation studies are insufficient. The authors may want to provide more experiments to show the effectiveness of the proposed BQ-based formulation.

**Summary Of The Paper:**

To improve the out-of-distribution generalization in solving combinatorial optimization (CO) problems, this paper proposes a transformed MDP to formulate constructive heuristics. Specifically, the transformation is based on bisimulation quotienting (BQ), which reduces the state space by leveraging the symmetries of the state space. Experiments on the Traveling Salesman Problems (TSP) and Capacitated Vehicle Routing Problems (CVRP) demonstrate that the proposed method achieves the state-of-the-art generalization performance.


**Summary Of The Review:**

The idea of improving the generalization performance based on BQ is interesting. However, some claims are not well-supported, and some description of the method is unclear.

---

> ### Author Response · Authors · 2022-11-18
> **Reply to reviewer kF44**
>
> Thank you for the constructive feedback. We address your comments below and we’ll be happy to clarify any further doubt you may have.
>
> **"The authors claim that their formulation is generic for arbitrary CO problems. However, the formulation seems specific for the TSP and CVRP. Thus, it would be more convincing if the authors could explain how the proposed formulation applies to other CO problems, such as mixed-integer linear programs."** We are not sure to understand which “formulation” you are referring to. Our claim is that given a CO problem, and a chosen specification, we provide a way to automatically derive a naive MDP for a constructive heuristic (such that solving the MDP is equivalent to solving the original CO problem). Then, if we can define a bisimulation $\Phi$ that exploits the symmetries of the CO problem, we provide a way to automatically derive the BQ-MDP induced by $\Phi$. We believe that the problem-specific bisimulations are often easy to find, even for non-routing problems (see General Reply).
>
> **"The details about introducing the bisimulation quotienting into their proposed transformer-based model are unclear."** Our proposed bisimulation quotienting allows to reduce the state space of the MDP. For the TSP (and CVRP), our neural architecture is essentially a plain Transformer encoder model. However our model takes as input the BQ-state (which does NOT include the partial solution), which makes it fundamentally different from previous works. Specifically, it does not have an encoder-decoder structure as the AM, POMO, MDAM, TransformerTSP models; it also has a $\mathcal{O}(N^3)$ complexity (versus $\mathcal{O}(N^2)$) that we hypothesize significantly helps with generalization. Indeed when we replace the quadratic attention by a linear equivalent, generalization performance significantly drops (cf Appendix C1 “Transformer VS HyperMixer as Model” ).
>
> **"please illustrate the proposed model architecture and hyperparameters."** Please see Sec 6 - “Hyperparameters and Training Procedure” for a precise description of the model’s architecture and hyperparameters. In addition, we have added a diagram for a quick visualization of our model (Figure 4 in the Appendix).
>
> **"The ablation studies are insufficient. The authors may want to provide more experiments to show the effectiveness of the proposed BQ-based formulation"** In order to do an ablation study about how the BQ-based state representation enables the performance and generalization of our model, we would need to apply the same model on the naive MDP state representation. But in such a state representation, we need to input the 'path-so-far', and therefore also modify the model (for example with positional encodings) to encode this path-so-far, additionally to remaining nodes. As a result, the model is changed, and the ablation is not very meaningful. However, we have provided 3 kinds of ablations for specific aspects of our model:
>
> 1. **NEW:** We have additionally tested an “approximate” version of our approach, where we don’t completely re-encode the remaining nodes from scratch at every step. Instead, we encode the graph once for the first L - 1 layers, and only recompute the last layer by masking out the visited nodes. This is similar to the Embedding Glimpse layer in MDAM [Xin et al 2021] and the Approximate Step-wise model in [Xin et al 2021b]. Note that this model is no longer based on the BQ-MDP, because visited nodes in the graph affect the outcome of the model, which the BQ-MDP prevents. While this approach accelerates the computation, it deteriorates the performance significantly: on TSP100 the optimality gap was 9.83% (versus 0.54% with the original model). This provides evidence for the effectiveness of our BQ-based formulation. (Details in Appendix D.2)
>
> 2. Replacing the self-attention layers with efficient linear layers. The deterioration of the results proved our point that the additional computation of the self-attention layers helps with the generalization performance (Appendix D.1).
>
> 3. Replacing the ReZero normalization by the usual BatchNormalization in Transformers – degrades performance and makes training unstable and more seed-sensitive. (Appendix D.3)
>
> **"The idea of improving the generalization performance based on BQ is interesting. However, some claims are not well-supported, and some description of the method is unclear."** We hope that we have clarified the claims in our response. Please let us know if we can further clarify any specific point about our method.

---

### Official Review · Reviewer_B3kw · 2022-11-04

**Confidence:** 2
**Correctness:** 3
**Technical Novelty And Significance:** 3
**Empirical Novelty And Significance:** 3
**Recommendation:** 8

**Clarity, Quality, Novelty And Reproducibility:**

### Clarity
- Maths and notations seem correct, but more detailed descriptions will make readers understand better. Since I was not familiar with bisimulation and bisimilarity, I had to spend a lot of time to understand those concepts although I believe authors try their best to describe those concepts.
  - e.g., in Section 3, it is mentioned that “bisimilarity is equivalently defined as the largest bisimulation (the union of all bisimulations)” but I couldn’t fully understand what this means.
  - In Figure 1, $\equiv_{\Phi}$ is used without definition.
  - When the step space is defined in Section 2, $\mathcal{T}^*$ is mentioned without definition.
- In Section 4, “hence removing the need for a separate decoder”; I guess this was stated since there’s no need to use autogressive decoder, but I believe FF network after Transformer encoding can be regarded as a decoder. The same question for Section 4, Summary; authors state “our model does not have a decoder”.
- In Section 4, “Note that (optimal) solutions are not directly in the form of trajectories”: can you please elaborate what it means?
- In Section 4, Complexity, it would be better to describe how O(N^3) was derived, why standard encoder-decoder spend O(t^2) in more detail.
### Quality
- I think most of contents are well-written.
- The results show the impressive empirical performance.
- In Section C, authors use efficient Transformer (HyperMixer) to reduce the complexity but show that this harms the generalizability. Can you please elaborate why HyperMixer results in poor generalization? If this is simply because of using Layer Norm and not using ReZero, are there any alternatives other than HyperMixer?
- At some point, it was unclear to me why training on small instances generalizes well to larger instances. For example, let’s consider generalization for TSP 1000 when we have a trained model on TSP 100. From the very beginning of evaluation to the point where less than 100 nodes remain, I believe it’s not possible to see any path-TSP states which were used to train the model. Then, what makes the model generalize well for larger instances?
### Novelty
- The formulation using bisimulation seems interesting and novel.
### Reproducibility
- The idea seems easily reproducible.


**Strength And Weaknesses:**

### Strength
- Constructive approach in CO and its relation to MDP are rigorously defined.
- Explanation on generalization based on bisimulation and bisimilarity is clearly stated.
- Strong generalization performance is shown through multiple experiments across two different CO problem instances (TSP, CVRP).
### Weaknesses
- Problems are restricted into the Euclidean settings. The algorithm’s applicability to non-Euclidean, non-symmetric COs is unclear.
- High computational complexity since pure Transformer is used. It was shown that using efficient Transformer techniques makes generalization performance poor.


**Summary Of The Paper:**

A step-wise neural CO algorithm is proposed, where the idea of bisimulation quotienting is used to define a state with the minimal but sufficient information. Authors mathematically formalize the solution-construction approach in CO problems and its MDP perspective (in Section 2). Then, the idea of bisimulation quotienting is introduced and how it is used to map TSP-MDP into path-TSP-MDP is discussed (in Section 3, with CVRP). With such formulation, a simple policy network that uses a Transformer (with slight modifications, e.g., removing PE) and is suitable for path-TSP-MDP (and its CVRP version) is proposed (Section 4), where the scalability of the model was empirically proven in a variety of experiments (Section 6).

**Summary Of The Review:**

The submission proposes an interesting symmetry perspective for constructive CO problems and shows impressive performance, which makes me vote for the acceptance of this work.

---

> ### Author Response · Authors · 2022-11-18
> **Reply to reviewer B3kw (Part 2/2)**
>
> **"In Section 4, Complexity, it would be better to describe how O(N^3) was derived, why standard encoder-decoder spend O(t^2) in more detail."**  We have added an explanation in Section 4 - Complexity. Essentially, because of the quadratic complexity of self-attention and the fact that we call our model N times with a graph of size t at each step, the complexity is proportional to $\sum_{t=1^N} t^2 = N(N+1)(2N+1)/6$ hence O(N^3). In the case of standard transformers, the encoder’s self-attention, decoder’s self-attention and decoder’s cross-attention contribute: $N^2 + \sum_{t=1}^N t^2 + \sum_{t=1}^N t(N-t)= N^2+N.N(N-1)/2$ hence $\mathcal{O}(N^3)$. However, efficient implementations (such that in [Bresson and Laurent 2021])] make the complexity of the decoder linear at each step, resulting in a total complexity of $\mathcal{O}(N^2)$. In [Kool et al 2019; Kwon et al 2020; Xin et al 2021] the AM decoder is used where there is no self-attention, but cross-attention between a context vector of fixed size and the $(N-t)$ remaining nodes. Therefore in these models, after one encoding of complexity $\mathcal{O}(N^2)$, at each step of the construction process, the decoder’s complexity is $\mathcal{O}(t)$ or $\mathcal{O}((N-t))$ for N-t nodes to visit. Whereas the complexity of our model at each step, for (N-t) remaining nodes, is $\mathcal{O}((N-t)^2)$. We have clarified this point in Sec 4 - Complexity.
>
> **"authors use efficient Transformer (HyperMixer) to reduce the complexity but show that this harms the generalizability. Can you please elaborate why HyperMixer results in poor generalization? If this is simply because of using Layer Norm and not using ReZero, are there any alternatives other than HyperMixer?**"  In Section 4, we hypothesize that a subproblem of size t (i.e., t nodes left to be visited) should receive a computation effort that grows in t. We think that larger complexity (like $\mathcal{O}(t^2)$ as in Transformers) is more appropriate than smaller complexity (like $\mathcal{O}(t)$ as in HyperMixer) for NP-hard problems like TSP and CVRP. The poor generalization results of HyperMixer confirm this. We believe that other linear-time models would yield similar results, although an exhaustive search among efficient Transformer alternatives is out of scope of this paper. LayerNorm instead of ReZero is not why HyperMixer generalizes poorly - we did try to use HyperMixer with ReZero at first, but found it to yield slightly worse results and less stable training.
>
> **"At some point, it was unclear to me why training on small instances generalizes well to larger instances. For example, let’s consider generalization for TSP 1000 when we have a trained model on TSP 100. From the very beginning of evaluation to the point where less than 100 nodes remain, I believe it’s not possible to see any path-TSP states which were used to train the model. Then, what makes the model generalize well for larger instances?"** We would like first to emphasis that the performance on TSP1000 of a model trained only on graphs with up to 100 nodes is precisely the kind out-of-distribution generalization (or extrapolation) that we target on the paper, i.e. we want a model that performs well on instances outside the training distribution (typically larger size, but also different spatial node distribution as in TSPlib). Our hypothesis is that our model generalizes well because of the use of the BQ-MDP that implies (i) having a reduced state-space and re-encoding the state at each step. Graphs in these states vary in size and distribution, implicitly encouraging the model to work well across sizes and node distributions, and generalize better than if such variations were not seen during the training. Note the similarity with the intention of the model by [Xin et al 2021b], except that this latter only approximates the re-embedding process in practice. We have added an experiment with an approximated version of our model (where at each step, the state embedding is only partially updated -- hence influenced by the past sequence) and the drop in performance confirms the effectiveness of the BQ-based formulation (see details in Appendix D2). And (ii) the inductive bias of our neural model that spends $\mathcal{O}(t^2)$ operations per step (cf ablation with HyperMixer).

---

> ### Author Response · Authors · 2022-11-18
> **Reply to reviewer B3kw (Part 1/2)**
>
> Thank you for the insightful comments and for taking the time to provide precise and constructive feedback. We address your comments below and we’ll be happy to clarify any further doubt you may have.
>
> **"Problems are restricted into the Euclidean settings. The algorithm’s applicability to non-Euclidean, non-symmetric COs is unclear."** Please note that our proposed bisimulations for TSP and CVRP would be valid for non-euclidian non-symmetric versions of these problems. The neural model would however be different, since the input would be a graph with features on the edges (the distances) and this is not directly handled by the Transformer architecture that we use. Another graph neural network such as Graph Convolutional Neural Networks [Kipf and Welling 2017] would be more appropriate. For applicability beyond routing problems, please see the general reply.
>
> **”High computational complexity since pure Transformer is used. It was shown that using efficient Transformer techniques makes generalization performance poor”** Most state-of-the-art NCO approaches use some variations of Transformer architectures (e.g. the cited TransformerTSP, AM and following works, also [Ma et al 2021]) or graph neural networks (GCN model [Joshi et al 2019], GIN in [Zhang et al 2020]) which have a quadratic complexity for fully connected graphs.  Given the NP-hardness of the considered problems, it is not surprising that performance deteriorates with less compute – in our case when using the HyperMixer that has linear complexity. Rather, we believe that the key question is where and how to spend the compute. In Section 4 paragraph Complexity, we argue that the BQ-transformation allows us to align the available compute with the difficulty of the subproblem, in contrast to previous approaches.
>
> **“it is mentioned that “bisimilarity is equivalently defined as the largest bisimulation (the union of all bisimulations)” but I couldn’t fully understand what this means”** We mean that if we denote by $\Psi$ the bisimilarity of an MDP, then whatever the bisimulation $\Phi$, and the states $s_1, s_2$,  $\Phi(s_1) = \Phi(s_2) \Rightarrow \Psi(s_1) = \Psi(s_2)$. We agree that this is not obvious and have added Appendix H to provide more background on bisimulation/bisimilarity for the interested readers – although it is not necessary to read it to understand the contribution of the paper.
>
> **"In Figure 1, $\equiv_{\boldsymbol{\Phi}}$ is used without definition"** Here $\equiv_{\boldsymbol{\Phi}}$ denotes the equivalence with respect to the mapping $\Phi$: $s_1 \equiv_{\boldsymbol{\Phi}} s_2$ iff $\Phi(s_1)=\Phi(s_2)$. We have added the definition in Sec 3 - Bisimulation Quotinting.
>
> **"$\mathcal{T}^\*$ is mentioned without definition"** We have added the definition: $T^*$ is the set of sequences of elements of $T$.
>
> **“hence removing the need for a separate decoder”; I guess this was stated since there’s no need to use autogressive decoder, but I believe FF network after Transformer encoding can be regarded as a decoder."** We agree that the FF layer on top of the final node embeddings can be viewed as a decoder. Our statement about not having a decoder refers to the fact that we do not have a separate module (with attention layers) after the encoding, that is called at each step of the construction process – like previous works that used attention-based architectures (e.g. AM, POMO, TranformerTSP). We have clarified this point in the manuscript (Sec 4 - Model/Summary).
>
> **““Note that (optimal) solutions are not directly in the form of trajectories”: can you please elaborate what it means?"** By trajectory we mean the sequence of construction steps. For the TSP, a solution (a tour) can be represented by as many different sequences as there are nodes, by varying the start node. Similarly for the CVRP, the solution being a set of subtours, any order of the subtours (and the clockwise or anticlockwise order of the nodes within each subtour) gives a different trajectory. A description of how we generate trajectories from the (ground-truth) solutions is presented in Appendix D.
>
> [Kipf and Welling 2017] Semi-Supervised Classification with Graph Convolutional Networks, ICML 2017\
> [Ma et al] Learning to iteratively solve routing problems with dual-aspect collaborative transformer, Neurips 2021 \
> [Joshi et al 2019] An Efficient Graph Convolutional Network Technique for the Travelling Salesman Problem, arXiv 2019\
> [Zhang et al 2020] Learning to Dispatch for Job Shop Scheduling via Deep Reinforcement Learning, NeurIPS 2020

---

### Author Response · Authors · 2022-11-18
**Reply to all the reviewers: Revised manuscript and highlighted changes**

We have uploaded a revised manuscript that includes many clarifications and the new KP experiments. We have also uploaded a diff file where the changes are highlighted for a quick overview.

---

### Author Response · Authors · 2022-11-18
**Reply to all the reviewers: Applicability beyond routing problems**

We thank the reviewers for their insightful comments and constructive feedback. We are glad that they found our **idea of using bisimulation quotienting in NCO novel and interesting** [B3kw, kF44, Sx6V] and that they found the **generalization performance of our model to large TSP and CVRP instances impressive** [B3kw, Sx6V] and **significantly better than current ML-based approaches** [kF44].

A common concern among the reviewers was the applicability and relevance of our contributions beyond the presented euclidian TSP and CVRP.
* B3kw: *"Problems are restricted into the Euclidean settings. The algorithm’s applicability to non-Euclidean, non-symmetric COs is unclear"*.
* kF44 *"The authors claim that their formulation is generic for arbitrary CO problems. However, the formulation seems specific for the TSP and CVRP. Thus, it would be more convincing if the authors could explain how the proposed formulation applies to other CO problems, such as mixed-integer linear programs"*.
* Sx6V: *"While the approach is presented as a generic and flexible framework for combinatorial optimization, it seems to require a domain-specific mapping (Phi). The paper only considers such mapping for two routing problems and it is not clear if such mappings would naturally exist for many other (particularly non-routing) combinatorial problems and whether they will prove useful in such problems"*.

To address this concern, we present here how our framework can be applied to 2 classical non-routing CO problems: the Knapsack Problem (KP) and Minimum Vertex Covering Problem (MVCP). We have also adapted and trained our neural model for the BQ-reduced KP MDP and obtained consistent improvements in generalization over a state-of-the-art baseline.

## Bisimulation Quotienting for the Knapsack Problem:
In the KP, given set $S$ of items, each with a value $v_i$ and weight $w_i$ and a knapsack of capacity $c$, the goal is to select a subset $S’ \subseteq S$ of items such that their total weight does not exceed the capacity and their total value is maximal. A solution of the KP can be constructed sequentially by selecting one item to add at each step. The partial solution at step $k$ has the form $s=(S, c, (i_1, …, i_k))$ where $i_j$ is the item packed at step $j$. The naive MDP state is given by the partial solution. We can design the bisimulation $\Phi$ that maps a partial solution to a new KP instance $(S_k, c_k)$ with $S_k=S$ \\ {$i_1, …, i_k$} and $c_k = c - (w_{i_1} + .. + w_{i_k})$. The MDP that results from quotienting the naive MDP by the bisimulation $\Phi$ (that we call BQ-KP) has a clearly smaller state space, while states preserve the minimal amount of information needed for the next decisions. With minor modifications of our path-TSP neural model, we have trained a policy network for BQ-KP and report a summary of the results:
|Problem      | POMO (single)    |  POMO (all)   | BQ (greedy)
|------------|-----------------:|--------------:|--------------:
|N=200, C=25  |       0.476%     |      **0.017%**   |      0.081%
|N=200, C=50  |       1.544%     |      1.170%   |      **0.056%**
|N=500, C=25  |       6.217%     |      4.897%   |      **0.904%**
|N=500, C=50  |       0.291%     |      **0.042%**   |      0.739%
|N=1000, C=25 |       5.386%     |      **3.973%**   |      5.808%
|N=1000, C=50 |       6.545%     |      5.724%   |      **4.267%**
|All          |       3.552%     |      2.648%   |    **1.980%**

We compare our results to the POMO model which was also applied to the KP. We use similar datasets and settings, and report the generalization performance of both models trained on KP200-C=25. We observe that similarly as for the TSP and CVRP, we obtain excellent results: with a simple greedy rollout of our model, we often outperform POMO (all) that returns the best of $N$ trajectories. A complete description of KP specification, the BQ-KP MDP, the adapted model and the experimental evaluation are presented in Appendix B.

## Bisimulation Quotienting for the Minimum Vertex Covering Problem

Given a graph $G=(V, E)$, the goal of the MVCP is to find the smallest subset of vertices $V’ \subseteq V$ such that every edge of the graph has at least one extremity in $V’$. A natural way to construct a solution is to select the vertices of V’ sequentially, until every edge is covered. Therefore a partial solution would have the form $s=(G, (v_1, …, v_k))$, where $(v_1, …, v_k)$ is the sequence of already selected vertices. This partial solution is the state of the naive MDP at step $k$. We can define the bisimulation $\Phi$ that maps this naive state to a subgraph where we remove the selected nodes and covered edges, i.e. $\Phi(s) = G’$, where $G’=(V’,E’)$, with $V’ = V$ \\ {$v_1$, …, $v_k$} and $E’=E$ \\{the edges that have at least one end point in {$v_1$, .. , $v_k$}}. Again, this bisimulation clearly reduces the state space by extracting the minimal information that is sufficient to select the next vertices.

---

### Decision · Program_Chairs · 2023-01-20

**Decision:**

Reject

**Justification For Why Not Higher Score:**

There was concensus among the reviewers in the written discussion for rejection.

**Justification For Why Not Lower Score:**

N/A

**Metareview: Summary, Strengths And Weaknesses:**

The paper presents a very interesting approach to leverage symmetries in combinatorial optimization (CO) problems.  While the approach advances the state of the art for solving CO by MDPs, the reviewers expressed concerns about the generality of the approach and the clarity of the technical details.  The rebuttal and the new experiments helped, but the concerns remain.  At the end of the day, the class of CO problems for which the approach is applicable should be clearly stated instead of broadly claiming that it applies to all CO problems.  In addition, the approach relies on a user defined mapping $\Phi$ and it is not clear how to obtain this mapping in general.  Finally, as pointed out by the reviewers, the technical details of the approach were not clear and the absence of code did not help.  Since the core of the work is good and worth publishing eventually, the authors are encouraged to pursue this work and to re-submit at a future venue while taking into account the reviewers' feedback.

Strengths:
* New MDP formulation that leverages symmetries induced by bisimilarity
* The proposed approach advances the state of the art

Weaknesses:
* Not clear what class of combinatorial optimization problems this approach is suitable for
* Users must provide a mapping $\Phi$, but it is not clear how to define it beyond the examples provided in the paper
* Lack of technical details regarding the implementation of the approach (this was not helped by the absence of code)